

# Investigating the Impacts of Saharan Dust on Tropical Deep Convection Using Spectral Bin Microphysics

Matthew Gibbons[1], Qilong Min[1], Jiwen Fan[2]

[1] Atmospheric Science Research Center, State University of New York, Albany NY 12203, USA
[2] Earth Systems Analysis & Modeling, Pacific Northwest National Laboratory

*Correspondence to*: Qilong Min (qmin@albany.edu)

**Abstract.** To better understand the impacts of dust aerosols on deep convective cloud (DCC) systems reported by previous observational studies, a case study in the tropical eastern Atlantic was investigated using the Weather Research and Forecasting (WRF) model coupled with a Spectral Bin Microphysics (SBM) model. A detailed set of ice nucleation
parameterizations linking ice formation with aerosol particles have been implemented in the SBM. Increasing IN concentration in the dust cases results in the formation of more numerous small ice particles in the heterogeneous nucleation regime (between -5°C and -38°C) compared to the background (Clean) case. Convective updrafts are invigorated by increased latent heat release due to depositional growth and riming of these more numerous particles, which results in increased overshooting and higher convective core top heights. Competition between the more numerous particles for
available water vapour during diffusional growth and available smaller crystals/drops during collection reduces particle growth rates and shifts precipitation formation to higher altitudes in the heterogeneous nucleation regime. Homogeneous ice formation is reduced in the dust cases as IN concentration is increased, due to more liquid drops converting to ice by freezing or riming before reaching -38°C and reduced peak supersaturation values from increased diffusional growth. Local IN activation in the stratiform regime contributes to increased cloudiness in the heterogeneous nucleation regime. A greater
number of large snow particles form in the dust cases, which are transported from the core into the stratiform regime and sediment out quickly. Together with reduced homogeneous ice formation, fewer particles form within and/or are transported into the anvil regime. This shifts the stratiform/anvil cloud occurrence frequency to warmer temperatures and reduces anvil cloud extents. Total surface precipitation accumulation is reduced proportionally as IN concentration is increased, due to less efficient graupel formation reducing convective rain rates. Stratiform precipitation accumulation is increased due to greater
snow formation and growth, but does not counteract the reduced convective accumulation. Riming efficiency in the dust cases is reduced due to smaller cloud ice crystals, resulting in smaller graupel sizes overall. Ice particle aggregation occurs earlier in the simulation and over a wider temperature range in the dust cases, which increases snow formation in the heterogeneous nucleation regime. Radar reflectivity values are increased in the dust cases at temperatures below 0°C in both the convective and stratiform regimes due to more large snow particles. More numerous small ice/graupel particles that form
in the heterogeneous nucleation regime in the dust cases melt and reduce reflectivity values in the convective core near the surface, consistent with case study observations.



## 1 Introduction

Deep convective clouds (DCC) are important sources of precipitation and play a strong role in both regional and global circulation, with tropical convection being particularly significant (Arakawa, 2004). The strong updrafts within convective clouds can transport small cloud particles to the level of neutral buoyancy where they spread out to form the anvil cloud

associated with DCC (Folkins, 2002; Mullendore, 2005). The greater area coverage and lifetime persistence of the anvil cloud compared to the convective core makes the anvil cloud important to global energy balance and radiative transfer. This makes the study of deep convective clouds important for current and future climate research (Solomon et al., 2007; Rosenfeld et al., 2013). Convective intensity is the primary determiner of the depth, area, and lifetime of the resulting anvil clouds (Futyan and Del Genio, 2007). However, observational and numerical studies of aerosol indirect effects (AIE)

suggest that changes to cloud microphysical processes can significantly modulate these macrophysical qualities (Fan et al. 2007a, 2010a, 2013; Min et al, 2009; Koren et al. 2010a, 2010b; Li et al., 2011; Niu and Li, 2012; Storer et al., 2014; Saleeby et al., 2016).

Dust aerosols have been observed at significant concentrations even in remote locations far from their expected source

regions (Prospero, 1999). They are predominately composed of insoluble silicate particles (Lohmann, 2002) which have been established to act as effective ice nuclei (IN, Pruppacher and Klett, 1997; DeMott et al., 2003; Sassen et al., 2003; Boose et al., 2016) and/or cloud condensation nuclei (CCN; Twohy et al., 2009; Kumar et al., 2011; Karydis et al., 2013). The Saharan Air Layer (SAL; Prospero and Carlson, 1970; Carlson and Prospero, 1972) is an elevated layer of dry air between 850-500 hPa, often containing lofted dust particles. The SAL has been observed interacting with tropical cloud

systems, such as tropical cyclones and mesoscale convective systems (MCS), and may impact their intensity and evolution (Karyampudi and Carlson, 1988; Dunion and Velden, 2004; Evan et al., 2008; Min et al., 2009: Zhang et al. 2009; Braun 2010; Lau et al., 2010; Carrios and Cotton, 2011; Cotton et al., 2012; Braun et al., 2013). A trans-Atlantic dust outbreak of Saharan origin occurring 1-10 March 2004 (Morris et al., 2006) was subjected to a rigorous multi-sensor and multi-platform observational analysis (Min et al., 2009; Li et al., 2010; Min and Li, 2010; Li and Min, 2010; Min et al., 2014). The

interaction of this dust outbreak with a well-developed MCS resulted in strong effects on cloud microphysical processes. Small ice particles were abundant in the heterogeneous nucleation regime in the dusty region. The size spectrum of the vertical precipitation structures was shifted from heavy to light precipitation (Min et al., 2009; Li and Min, 2010). Substantial changes to cloud top distributions and precipitation profiles resulted from a change in the partition between homogeneous and heterogeneous ice formation processes under dusty conditions. Such macrophysical changes in the cloud

systems resulted in substantial thermal infrared radiation cooling of up to $16\,\mathrm{W\,m^{-2}}$ (Min and Li, 2010). The reported changes to cloud top distribution and the partition between homogeneous and heterogeneous ice formation differ from those described by studies focusing on the CCN activation of aerosols.





Observational and modeling studies of DCC have shown different results relating to the effect of aerosol on convection and precipitation, indicating that aerosol may either enhance or suppress convection and precipitation depending on aerosol concentration and environmental conditions (Khain and Pokrovsky, 2004; Khain et al., 2004, 2005, 2008; van den Heever et al., 2006; Fan et al., 2007b; Lee et al., 2008; Min et al., 2009; Min and Li, 2010; Li and Min, 2010; Min et al., 2014; Altaratz

et al., 2014). Clouds forming in elevated aerosol environments exhibit reduced cloud drop effective radii as a result of a greater number of smaller drops forming (Andreae et al., 2004; Koren et al., 2005). This can result in less efficient collision-coalescence processes (Khain et al., 2005) which shifts the formation of precipitation to higher altitudes in the clouds. Condensation and evaporation processes are affected by the altered drop size distribution and number concentration, resulting in changes to the location and intensity of latent heat release within the cloud (Khain et al., 2005; Rosenfeld et al.,

2008). The higher droplet concentrations induce greater condensation and latent heat release, resulting in stronger convective updrafts and the formation of taller and wider clouds (Frederick, 2006; Zhang et al., 2007). Increased evaporation of smaller drops can result in stronger cold pool formation and enhanced secondary convection (Khain 2009, Lee et al., 2010). Conversely, other studies have noted that the formation of larger drops due to enhanced rain drop collision-coalescence limits evaporation and weakens the cold pool (Altaratz et al. 2007; Berg et al. 2008; Lerach et al., 2008, Storer et al., 2010;

Lim et al., 2011; May et al., 2011, Morrison, 2012; Grant and Van Den Heever, 2015). Aerosol indirect effect related changes to cloud macrophysics are frequently attributed solely to convective invigoration by the increased liquid and/or ice particle number concentrations and subsequent changes to diffusional growth processes. However, a study by Fan et al. (2013) involving simulations of DCC in three different regions, suggested that the observed taller and wider clouds could be better explained by changes to microphysical properties such as the particle size distribution. Thermodynamic invigoration

by increased latent heat release did not unanimously occur in the study when polluted conditions were simulated, although increased cloud fraction and cloud top height were present. The study noted that the reduced hydrometeor sizes in the polluted case allowed greater cloud mass to be detrained from the convective core, and decreased particle fallout speed that slows down the cloud anvil dissipation.

Earlier numerical studies of aerosol-cloud interactions tend to focus upon the action of soluble aerosols as CCN, with changes to ice formation resulting from the affected liquid processes only (Khain et al., 2005; Fan et al., 2009b, 2012a, 2012b; Storer and van den Heever, 2013; Saleeby et al., 2016). However, DCC can also be sensitive to the aerosols that act as IN (Van den Heever, et al., 2006; Ekman et al., 2007; Tao et al., 2012). The study of Van den Heever et al. (2006) described the differing impacts of CCN and IN on convective clouds and subsequent anvil development. They found that

increasing CCN concentration tended to reduce surface precipitation. Increasing IN concentration initially increased surface precipitation, but eventually reduced the total to less than the control case by the end of the simulation. Updraft intensity increased with the increased aerosol concentration due to stronger latent heat release, but anvils were generally smaller and more organized. Ekman et al. (2007) studied the sensitivity of a continental storm to IN concentration and found that updrafts were enhanced due to added latent heat release from ice crystal depositional growth. The stronger updrafts enhanced



homogeneous nucleation, increasing anvil cloud coverage and precipitation. Fan et al. (2010a) compared the effects of CCN and IN on convection and precipitation and noted that the CCN effect is more evident in changing cloud anvil size, lifetime, and microphysical properties. IN was shown to have a small effect on convective strength, but the microphysical effects could still be significant. However it should be noted that Fan et al. (2010a) did not have a prognostic IN treatment as what we have done for this study.

Aerosols such as dust influence the character of individual clouds and storms, but evidence of a systematic effect on storm or precipitation intensity is still limited and ambiguous. Therefore detailed numerical models are required to understand the dynamical and microphysical changes that result in the observed effects of dust on DCC. However, the representation of DCC processes relevant to aerosol-cloud interactions is still considered weak, due to some of the fundamental details of cloud microphysical processes still being poorly understood. This is particularly true with regards to ice and mixed-phase clouds (Boucher et al., 2013). This low confidence is a result of the complex coupling between the processes controlling cloud and precipitation properties, which cover a wide range of spatial and temporal scales (Tao et al., 2012). Large uncertainties also exist in ice nucleation parameterizations within numerical models (DeMott et al., 2010). However, comparison of model results with a well observed case study, such as the multi-platform and multi-sensor Min et al. (2009) study, can limit the impact of these uncertainties when analysing results from numerical simulations. Ice formation in deep convective clouds may result from heterogeneous and/or homogeneous ice nucleation depending on the depth of the cloud and the chemical composition of the background aerosols. Heterogeneous ice nucleation can occur at temperatures between -5°C and -38°C via the mechanisms of deposition, immersion, and contact freezing (Vali et al., 1985; Vali et al., 2015) when ice nuclei (IN) are present. Homogeneous ice nucleation involves droplet and aerosol haze particle freezing at temperatures lower than -38°C (Koop et al., 2000; Mohler et al., 2003; Ren and MacKenzie, 2005). Deep convection frequently shoots liquid drops up to the upper troposphere where the temperature is colder than -38°C, leading to strong homogenous droplet freezing. Therefore, a comprehensive handling both heterogeneous and homogeneous ice formation mechanisms must be incorporated into numerical simulations to gain a clearer understanding of ice formation in DCC.

Observations suggest that the presence of IN particles such as dust has a significant impact on the microphysical and macrophysical properties of DCC, but many numerical simulations rely on a relatively simple handling of IN particles and the associated heterogeneous ice formation mechanisms. Accurate simulations of ice formation processes in DCC require ice nucleation to be directly linked with IN concentration. In this study, we add a prognostic IN variable to allow for the transport of IN particles by the wind field and the removal of IN by heterogeneous ice formation. We also update the set of heterogeneous and homogenous ice nucleation parameterizations within the WRF-SBM to connect ice nucleation with dust particles. Heterogeneous ice formation resulting from the updated immersion, contact, and deposition-condensation freezing schemes account for the full range of ice formation mechanisms active at temperatures between -5°C and -38°C. Detailed information on specific updates made to the model has been provided in section 2. We simulate the observed MCS occurring



on 08 March 2004 in the tropical eastern Atlantic under background (Clean) and dust affected conditions. The Clean case will be used as a baseline to evaluate the model's skill at reproducing the observed cloud and precipitation fields of DCC not affected by the observed dust outbreak. The dust cases will test the sensitivity of the baseline case to different number concentrations of IN. Comparing the changes experienced by the dust cases with observations will allow us to test the

sensitivity of various ice formation mechanisms within the MCS to the presence of dust and verify the hypotheses of Min et al. (2009) and the later associated studies. Radar reflectivity measurements provide a valuable insight into the microphysical impacts of aerosols, such as dust, on DCC when analysed in conjunction with detailed numerical simulation results. However, radar reflectivity is sensitive to the number concentration, PSD, phase, density, fall rate, and spatial orientation of precipitation particles (Ryzhkov et al., 2011). These qualities are difficult to track accurately when a numerical model relies

on the fixed PSDs frequently used within bulk microphysics schemes. The use of bin microphysics allows for explicit calculation of microphysical processes that affect cloud and precipitation formation and growth. In addition, the bin PSDs can be directly converted into radar reflectivity values that can be compared with observations. Where appropriate, we have separated results into convective and stratiform clouds to address the distinct microphysical and macrophysical changes occurring within those cloud regimes.

## 2 Model Description

Numerical simulations were undertaken using the WRF version 3.1.1 developed by the National Center for Atmospheric Research (NCAR) as described in Skamarock et al. (2008). WRF solves the fully compressible, non-hydrostatic Euler equations formulated on terrain following hydrostatic-pressure coordinates and the Arakawa C-grid. The model uses Runge-Kutta second- to sixth-order advection schemes in both horizontal and vertical directions. The fifth-order advection scheme

is used in this study. The monotonic technique is employed for advection of scalar and moist variables. The cloud microphysical scheme is described below.

### 2.1 Spectral Bin Microphysics (SBM)

The original SBM (Khain et al., 2004) solves a system of kinetic equations for the size distribution functions for 7 hydrometeor types: water droplets, ice crystals (plate, column, and dendrite), aggregates, graupel, and frozen drops/hail. An 8th size distribution function exists for CCN. Each size distribution is represented by 33 mass doubling bins, where the mass of a particle in each bin is twice the mass of a particle in the preceding bin. A fast version of the SBM (Fast-SBM) with four size distributions of water drops, low density ice (ice crystals and aggregates), high density ice (graupel and hail), and

aerosol (CCN) was created in order to substantially reduce the computational costs (Khain et al. 2009, Fan et al 2012a) and is the version used in this study. Further details about the mechanics of the SBM are found in Khain et al. (2004) and Fan et al. (2012a) and will not be repeated here.



In order to examine IN impacts on clouds and precipitation, an additional prognostic variable for IN particle (dust in this case) number concentration was added to the model as detailed in Fan et al. (2014). We update the heterogeneous ice nucleation parameterizations in the SBM (as detailed in the following section) to connect ice formation with dust particle concentrations. In this study, a dust layer located between 1 and 3km has been added to the dust case simulations, to reproduce a similar dust layer present in the observed case. The dust layer is initialized to cover the entire 4[th] domain at model start-up and thereafter is resupplied exclusively from the lateral boundaries of the 4[th] domain by wind advection. The dust in the layer can serve as IN, CCN, or some fractional combination of the two by means of a simple partition which is set by the user depending on assumed or measured particle chemistry. This allows us to test the sensitivity of clouds within our model to a mixture of nuclei. We have set the dust layer to be IN exclusively in this study. Therefore, these dust cases will represent the maximum potential effects on heterogeneous ice formation for a given dust number concentration. Additional information on the CCN and IN number concentration values used in this study is provided in section 3.

## 2.2 Ice Formation Parameterizations

The original SBM (Khain et al. 2004) included both homogeneous and heterogeneous ice formation, but did not directly connect ice formation to a prognostic IN variable. Liquid drop freezing for both homogeneous and immersion mechanisms was provided by Bigg (1953). Ice formation resulting from condensation and deposition freezing was provided by Meyers et al. (1992). Contact freezing was not included in the original SBM. In order to perform a study of aerosol impacts on heterogeneous ice formation, it is necessary to directly link ice nucleation rates to aerosol properties. The study of Gong et al. (2010), and more recently Fan et al. (2014), updated the available homogeneous freezing mechanisms and additionally implemented separate parameterizations into the SBM for depositional, contact and immersion freezing, with ice formation in each of these schemes directly linked to the prognostic IN variable. In this study, we followed the Bigg (1953) for homogeneous freezing of drops. The heterogeneous ice nucleation parameterizations employed are detailed as below.

### 2.2.1 Heterogeneous Ice Nucleation and Freezing Schemes

Currently there is no deposition and condensational nucleation parameterization connecting with aerosol properties and developed based on deep convective clouds. As noted in Meyers et al. (1992) it is difficult to distinguish the relative contributions of depositional and condensational freezing in a parcel, since both form similarly sized small ice crystals, despite the different mechanisms of vapour to ice in the former and condensation followed immediately by freezing in the latter case. However, studies suggest that small ice crystals formed in the -5°C to -10°C temperature range can have a large impact on subsequent ice formation at higher altitudes (Ackerman et al. 2015; Hiron and Flossman, 2015; Lawson et al., 2015). A depositional-condensational scheme would allow for these small ice crystals to form in this specific temperature





range. To link depositional and condensational freezing with aerosols, we follow the implementation of van den Heever et al. (2006), updated from the Meyers et al. (1992) parameterization. The number of ice crystals generated by depositional-condensational nucleation ($N_{dep}$) is proportional to the IN number concentration ($N_{IN}$; $l^{-1}$) within the grid cell by Eq. (1):

$$N_{dep} = N_{IN}F_M, \tag{1}$$

where $F_M$ (unitless) is the function of the depositional-condensational nucleation by Meyers et al. (1992) that represents the fraction of the maximum available IN ($N_{id}$; $l^{-1}$) concentration that may be activated for the given conditions as calculated in Eq. (2):

$$N_{id} = \exp\{-6.39 + 0.1296[100(S_i - 1)]\}, \tag{2}$$

with $S_i$ being the saturation over ice. The value of $F_M$ is equal to 1 for conditions at ice supersaturation of 40%, at which point all IN are activated, and is equal to 0 when supersaturation over ice is negative. The initial size of an ice crystal formed by this scheme is assumed to be 2.5 μm in radius and is assigned to the smallest ice size bin.

As stated above, the immersion freezing mechanism in the original SBM uses the parameterization of Bigg (1953), which is temperature-dependent only. To provide an aerosol-based immersion freezing scheme, we have incorporated the parameterization of DeMott et al. (2015), which was implemented by Fan et al. (2014) (cited as DeMott et al. (2013) in Fan et al. (2014) due to DeMott et al. (2015) not yet being published). The DeMott et al. (2015) immersion freezing rate is parameterized as in Eq. (3):

$$N_{imm} = (CF)(N_{IN})^{(\alpha(273.16-T_k)+\beta)}\exp(\gamma(273.16-T_k)+\delta) \tag{3}$$

CF is an instrumental correction factor with a value of 3. Coefficients α, β, γ, and δ are 5.95E-5, 1.25, 0.46, and -11.6, respectively, representing mineral dust particles (DeMott et al., 2015) $T_k$ is the cloud temperature in degrees Kelvin, $N_{IN}$ is the number concentration of total aerosol particles with diameter larger than 0.5 μm, and $N_{imm}$ is the maximum number of immersion ice possible in the given temperature range. Liquid drops are consumed over the size spectrum starting with the largest sizes down to the smallest until the minimum of $N_{imm}$ or drop number is reached. According to Yin et al. (2005), drops with a radius smaller than 79.37 μm will be frozen to pristine ice crystals, otherwise graupel is formed.

We have also adopted the contact freezing parameterization of Muhlbauer and Lohmann (2009), which is based on Cotton et al. (1986) and Young (1977). In this parameterization, contact freezing is a result of the collision of supercooled liquid water drops and IN due to Brownian motion. The contact freezing rate is therefore proportional to the drops' radius and number



concentration. It is also proportional to the IN number concentration and Brownian diffusivity in air. Unlike Muhlbauer and Lohmann (2009) who calculated the freezing rate for the sum of all drops, we perform the calculation in this study for each spectral bin of drops. Then, the contact freezing rate ($N_{cnt}$; $l^{-1}s^{-1}$) for each individual size bin is represented by Eq. (4):

5    $$N_{cnt} = 4\pi r_c N_c D_k N_{IN},\tag{4}$$

where $r_C$ (m) and $N_C$ (m-3) is radius and number concentration of drops in the individual size bin, respectively. $D_k$ is the dust aerosol Brownian diffusivity ($m^2s^{-1}$), and is parameterized by Eq. (5):

10    $$D_k = \frac{k_B TC}{6\pi\eta r},\tag{5}$$

$D_k$ is a function of the Boltzmann constant $K_B=1.28 \times 10^{-23}$ $m^2$ kg $s^{-2}$ $K^{-1}$, T is the air temperature, r is the dry dust aerosol median radius, $\eta$ is the viscosity of air and C is the Cunningham slip correction factor. The viscosity of air depends on temperature, as calculated by Eq. (6):

$$\eta = 10^{-5}[1.718 + 4.9 \times 10^{-3}(T - 273.15) - 1.2 \times 10^{-23}(T - 273.15)^2],\tag{6}$$

The Cunningham slip correction factor is calculated by Eq. (7):

20    $$C = 1 + 1.26\frac{\lambda}{r}\frac{1013.25}{p}\frac{T}{273.15},\tag{7}$$

with the molecular mean free path length of air $\lambda=0.066$ μm, r is the dry aerosol radii, and p is the pressure. To simplify the calculation, the contact freezing number is the available dust number concentration $N_{IN}$, with freezing efficiency of 1. Upon freezing, drops with a radius smaller than 79.37 μm will be frozen to pristine ice crystals, larger drops will be frozen as 25    graupel.

It should be noted that currently there is no ice nucleation parameterization specifically developed for DCC, and the understanding of ice nucleation for DCC is still very limited. The best we can do for model simulations at this time is to employ the currently-available ice nucleation parameterizations for connecting with dust particles, evaluate our baseline 30    simulation with observations, and carry out model sensitivity tests based on the validated case simulation to understand the dust impacts and associated mechanisms.

**2.3 Radar Reflectivity Calculations**





The liquid and frozen hydrometeor PSDs calculated by SBM can be easily converted into radar reflectivity values, providing a bridge for the comparison of model simulated microphysical parameters with observable variables. For our study, we calculate radar reflectivity directly from the model's PSD for each of the individual hydrometeor species using the spherical

particle approximations of the Rayleigh scattering equations suggested by Ryzhkov et al. (2011). Reflectivity is calculated for each bin and then summed over the entire PSD to obtain the total for each hydrometeor species (rain; snow; graupel) which are then combined to obtain the total reflectivity. The general equation for snow and graupel reflectivity is represented by Eq. 1:

$$Z = \left(\frac{\rho_{s,g}}{\rho_i}\right)^2 \frac{|Ki|^2}{|Kw|^2} \int_0^\infty D^6 N(D)\, dD \qquad (1)$$

Where N(D) is the number concentration per cubic meter of snow(graupel) particles of Diameter (D) in millimetres. Density of snow or graupel is represented by $\rho_{s,g}$, while $\rho_i$ is the density of solid ice. $|K_i|^2$ and $|K_l|^2$ represent the dielectric factors of solid ice and liquid water, respectively. When calculating the reflectivity for liquid drops, the two leading ratios are equal to

1, but otherwise the equation is the same. The density relationship in the leading ratios can be expanded and simplified into a constant times the snow(graupel)-liquid density ratio, following Smith (1984) and Fovell and Ogura (1988) as in Eq. 2:

$$\left(\frac{\rho_{s,g}}{\rho_i}\right)^2 \frac{|Ki|^2}{|Kw|^2} = \left(\frac{\rho_{s,g}}{\rho_l}\right)^2 \left(\frac{\rho_l}{\rho_i}\right)^2 \frac{|Ki|^2}{|Kw|^2} = 0.224 \left(\frac{\rho_{s,g}}{\rho_l}\right)^2 \qquad (2)$$

Where $\rho_l$ represents the density of liquid. This is then substituted into Eq. 1 to yield Eq. 3:

$$Z = 0.224 \left(\frac{\rho_{s,g}}{\rho_l}\right)^2 \int_0^\infty D^6 N(D)\, dD \qquad (3)$$

The reflectivity values calculated for liquid drops, snow and graupel are then added together to obtain the total reflectivity,

which is converted to dBZ by Eq. 4:

$$Z_{dBZ} = 10\, Log(Z_{total}) \qquad (4)$$





## 3 Experiment Design

In our study, we have conducted experiments simulating the 08 March 2004 MCS described in Min et al. (2009), using realistic initial and boundary conditions. Four one-way nested domains were used (Figure 1), with horizontal grid resolutions of 81km, 27km, 9km, and 3km respectively and 41 vertical levels in each domain. Vertical level grid spacing is coarsest

(~800m) at the top of the atmosphere, becoming progressively finer near the surface to a minimum of ~30m. The numbers of horizontal grid points in each domain are 81x81, 81x81, 81x81, and 150x150, respectively. Initial and boundary conditions for the first domain are provided by the 1° x 1° 6-hourly National Centers for Environmental Prediction (NCEP) global final analysis dataset, with initial conditions for the other three domains being interpolated from the first domain. Due to the SBM not being designed to run at coarse resolutions, the SBM provides microphysics for only the 3km resolution domain with

bulk microphysics being selected for domains 1-3. The specific WRF parameterizations selected for the experiments are detailed in Table 1. Each case was run for 33 model hours, beginning at 18Z 07 March 2004.

The initial number concentrations of CCN are kept identical between the different cases. Typical marine aerosol number concentrations tend to be low, on the order of 300-600 cm$^{-3}$ (O'Dowd et al., 1997; Yoon et al., 2007). Therefore, the CCN

number is set to a uniform value of 300 cm$^{-3}$ below 2km with the CCN number being reduced exponentially from this value as height increases above 2km. The initial IN distribution is set to be vertically uniform at .01 cm$^{-3}$ for the Clean case. The dust cases add an increasing number of IN to the Clean case's background value in a layer located vertically between 1km and 3km, as described by Min et al (2009). The dust layer contributes IN to the smallest domain only, as the bulk microphysics used in the larger domains do not directly connect dust with ice formation. The dust cases are set with different

IN numbers within the dust layer of 0.12 cm$^{-3}$ (case D.12), 1.2 cm$^{-3}$ (case D1.2), and 12 cm$^{-3}$ (case D12), respectively. These values were selected based on aerosol measurements (Table 2) that were taken during the trans-Atlantic Aerosol and Ocean Science Expeditions (AEROSE) experiment (Morris et al., 2006) for dates coinciding with the observational study of the March 2004 dust outbreak detailed in Min et al. (2009). The dust loading was assumed to be the difference in the aerosol number of the dusty and pristine periods. Only aerosol particles with a radius greater than 0.5 microns were considered when

taking this difference, due to the smaller aerosol sizes being more prevalent during the pristine period compared to the dusty period. This size range is consistent with the study of DeMott et al. (2015) for ice nucleating particles. The resulting dust number was multiplied by an activation fraction suggested by Niemand et al. (2012) for Saharan dust to arrive at the number used for case D.12. Other studies have suggested that dust related IN numbers greater than 1.0 cm$^{-3}$ are possible (DeMott et al., 2003; Sassen et al., 2003; Ansmann et al., 2008), so two additional dust cases with IN numbers one (D1.2) and two (D12)

orders of magnitude greater than the initial D.12 case were included in the study.

To prevent the CCN and IN fields from being diluted due to the inflow of air from the lateral boundaries, the CCN and IN numbers of the outer five grid cells (i.e., the boundary points) on each side of domain 4 are set to the initial values



throughout the integration period. The initial vertical profile of domain averaged relative humidity shows moist (>60% RH) air below 6km and drier air (<50% RH) above 6km, while the profiles of horizontal winds evidence weak (<5 m/s) to relatively weak (<10m/s) wind shear below 7 km, following the criteria used by Fan et al. (2009b). After the model's 6 hour spin up time, a relatively dry air layer corresponding to the SAL enters the domain via the NCEP-FNL boundary conditions

and is present for the duration of the simulation.

Additional criteria used to select subsets of the data for the purpose of our analysis are as follows. Cloudiness within an individual 3D grid cell was determined by the sum of all condensates within it exceeding a $10^{-6}$ kg/kg threshold value, following the definition used in Fan et al. (2013). Cloud top was determined, from the top level of the model down to the

surface, as the highest level with at least two consecutive levels exceeding the cloudiness threshold, which was intended to limit the influence of very thin clouds on the resulting analysis. While this does not take multiple cloud layers into account, it is similar to the top-down view of clouds observed by many satellites. To sort results by precipitation regime, we adapt the definitions of Fan et al. (2013) for convective and stratiform precipitation, with each vertical column classified as a single precipitation type only. For all precipitating clouds, surface rain rates must exceed 0.05 mm/hr. Convective precipitation is

classified as precipitating column with vertical motion exceeding a 1m/s threshold and cloud thickness of 8 km or greater. Non-convective precipitating columns are classified as stratiform by the presence of ice-phase precipitation in the column. Non-precipitating columns with a cloud layer thicker than 1km and both cloud top and cloud bottom temperatures colder than 0°C are classified as anvil clouds. Precipitating columns with cloud top temperatures warmer than freezing are classified as rain producing warm clouds.

**4 Results**

Min et al. (2009) reported a unique case of a mature MCS partially under the effects of a Saharan dust outbreak. They noted distinct changes to cloud microphysical and macrophysical properties when comparing the dusty and dust-free sectors of the MCS. Large-scale meteorological conditions drive the initial cloud formation and growth processes which are then modulated by aerosol indirect effects on cloud microphysical processes. Figure 1 describes the locations of the four model

domains, displaying the Atmospheric Infrared Sounder (AIRS) retrieval (Figure 1a) and the domain 1 model simulated precipitable water averaged over the duration of the simulation (Figure 1b). The large scale patterns of precipitable water are well reproduced by the model, although we note that the magnitude is slightly overestimated over the African continent and underestimated over the southern Atlantic compared to observations. Despite this, the magnitude in the location of our smallest domain is well reproduced, suggesting that the meteorological conditions in our region of interest are represented

sufficiently well.

**4.1 Microphysical and Macrophysical Changes**





Increasing IN concentration in the dust cases results in greater ice formation and growth within the heterogeneous nucleation regime. This affects homogeneous ice formation by reducing the number of liquid drops that reach the -38°C threshold and also by reduced peak supersaturation values due to the growth of more numerous ice particles within the heterogeneous nucleation regime. Figures 2 and 3 depict the vertical cross-section of a specific convective core and its associated stratiform/anvil cloud at a single model time step (hour 15) from the Clean and D1.2 cases. The cross-section slices are not identically located in the two cases due to small differences in the spatial evolution of the system, but are less than 3 grid points apart. In both cases, the slices are similarly located within their respective cloud system and are at similar stages of evolution. The slices are averaged zonally over 9km to further reduce the effects of spatial variations. The Black and dashed blue lines (Figure 2; Figure 3) depict updrafts (> 1m/s) and downdrafts (< -0.1 m/s). The grey dashed line (Figure 2; Figure 3) depicts the threshold value of cloudiness suggested by Fan et al. (2013) and shows the change to cloud geometry directly. The Clean case (Figure 2a) shows the classic DCC cloud structure of convective core and associated stratiform region transitioning into the anvil. The D1.2 case also possesses a similar cloud structure, but with a far smaller anvil cloud, which is a result of the changes to the partition between homogeneous and heterogeneous ice formation in the D1.2 case (Figure 7). Cloud formation is increased in the heterogeneous nucleation regime (Figure 2d) compared to the Clean case (Figure 2a). Liquid drops that would otherwise freeze homogeneously at temperatures colder than -38°C are converted to ice at warmer temperatures due to increased riming and/or immersion/contact nucleation. In addition, increased ice formation and growth within the heterogeneous nucleation regime reduces peak supersaturation values at colder temperatures, limiting ice formation in the homogeneous regime. Therefore, fewer particles form within and/or are transported into the anvil regime which limits its horizontal extent compared to the Clean case.

The first column of Figure 2 describes total water content (TWC), while columns 2 and 3 describe rain rate and radar reflectivity, respectively. TWC is increased in the dust case (Figure 2d) at temperatures below 0°C compared to the Clean case (Figure 2a). The higher TWC in the heterogeneous nucleation regime is accompanied by a correspondingly larger area of strong (> 1m/s) vertical motion. This supports the evidence of convective invigoration due to increased latent heat release in the dust-affected deep, high-IWP clouds reported by Min and Li (2010). Stronger updrafts in the dust cases supply sufficient water vapour to support the formation and growth of more numerous particles in the heterogeneous nucleation regime and can transport a greater number of large particles to higher altitudes in the convective core and into the adjoining stratiform regime. These large particles contribute to the higher rain rate values noted in the D1.2 case (Figure 2e) compared to the Clean case. The increased rain rates at temperatures below 0°C also correspond to the increased radar reflectivity values in the stratiform regime from the convective core almost to the anvil regime near the equator (Figure 2f). Figure 3 describes the effective radii (Re; 1.e2 um) of rain drops, graupel, and snow particles in columns 1-3, respectively. Rain drop radii are significantly decreased in the heterogeneous nucleation regime due to large sized drops freezing by





immersion/contact nucleation or by collisions with ice particles (riming; Figure 10) leaving smaller drops unfrozen. Graupel and snow radii are both decreased at temperatures below 0°C. This reduction is most pronounced within the convective core where competition between more numerous small particles during collision-collection reduces growth rates (Figure 9). At temperatures above 0°C, graupel radii is increased in the dust cases due to immersion freezing of large rain drops into

graupel within the heterogeneous nucleation regime and then falling into warmer temperatures. At temperatures below -38°C and in the anvil cloud regime, graupel and snow radii are increased compared to the Clean case. This is due to the stronger outflow in the dust cases from the convective core, which transports large precipitation particles greater distances before they sediment out of the cloud. In addition, precipitation formation is shifted to colder temperatures (higher altitudes) in the heterogeneous nucleation regime which increases the number of large particles forming near the cloud tops.

Aerosol indirect effects on cloud microphysical processes can result in a cloud top distribution that is higher or lower than would be expected for a given dynamical intensity. Figure 4 describes the changes to cloud top distribution in each of the three dust cases with respect to the Clean case. The cloud top distribution in Figure 4 combines all cloud types together to describe the overall macrophysical changes due to increasing IN concentration. We determine the cloud top by selecting the

highest vertical model level in each column that exceeds the 1.e-6 kg/kg cloudiness threshold value. While this does not take multiple cloud layers into account, it is similar to the top-down view of clouds observed by many satellites. Figure 4a describes the time series of cloud top occurrence frequency for the Clean case. The percentage at each model time represents the horizontal sum of all cloud tops occurring at a given model level divided by the total horizontal and vertical sum of cloud tops occurring at that specific model output time. Figure 4b through Figure 4d describes the dust case minus Clean case

difference of cloud top percentage. Increasing IN concentration from the D.12 case value in our simulations results in the overall cloud top height distribution shifting to lower altitudes (warmer temperatures). This is consistent with the findings of Min and Li (2010) in which higher AOD values were correlated with warmer cloud effective temperature. These macrophysical changes in cloud top distribution were noted to result in a strong cooling effect of thermal infrared radiation of up to 16 Wm$^{-2}$.

The cloud system transitions from shallow to deep convection between model hours 6 to 12. The majority of cloud tops occurring before hour 10 are warmer than -5°C. Therefore, the temperature and supersaturation conditions within these clouds are not sufficient for IN to activate and form ice crystals. Hence, the effects of increasing IN are limited during this time period. After the transition to deep convection, the cloud top distribution is shifted to lower altitudes (warmer

temperatures) between model hours 12 to 24. Cloud tops occur less frequently above 15km and more frequently between 12 and 13km as a result of the changes in the partition between homogeneous and heterogeneous ice formation. This is most pronounced in the D1.2 and D12 cases which both feature significant increases in heterogeneous ice formation compared to the Clean case. The numerous ice crystals that form when large concentrations of IN are activated compete for available water vapour during diffusional growth. The consumption of the cloud's available water vapour reduces peak supersaturation





at colder temperatures and suppresses homogeneous ice nucleation. We note that the shift in cloud top distribution is not linear with increasing IN number concentration. While both the D1.2 and D12 cases feature lowered clouds (hour 12-24), the differences in the D12 case are not as pronounced as in the D1.2 case. This is a result of greater concentrations of small cloud ice particles (Figure 9) in the D12 case compared to the D1.2 case. The small ice particles remain near the cloud top after larger particles sediment out, yielding a higher cloud top distribution relative to the D1.2 case. After model hour 24, the cloud top distribution is significantly lowered in the D12 case compared to the D1.2 case. The greater condensate mass of the D12 case allow more large snow particles to form (Figure 9d), which sediment out more quickly compared to the D1.2 case (Figure 9c). The small IN number in the D.12 case results in a cloud top distribution that is different from both the D1.2 and D12 cases. From model hours 20 onwards, The D.12-Clean case difference plot suggests that higher cloud tops are occurring compared to the other cases. However, average convective updrafts are slightly weaker during this time period compared to the Clean case (Figure 11a). This suggests that cloud microphysical changes are the cause of the higher clouds in the D.12 case. Specifically that particle sizes are smaller, allowing for increased vertical transport and slower sedimentation rates. The corresponding overall changes to cloud top height (averaged over model hours 6-33) are: Clean (12.64 km); D.12 (12.79 km, +1.14%); D1.2 (12.33 km, -2.49%); D12 (12.13 km, -4.08%).

## 4.2 Radar Reflectivity CFADs

With advances in observing technology, cloud and precipitation radars are used extensively for studying cloud and precipitation formation and microphysical-dynamical interactions. Min et al. (2009) used contoured frequency by altitude (CFAD) plots to describe the observed changes to convective and stratiform radar reflectivity between the dusty and dust-free regions. They noted that radar reflectivity at temperatures above 0°C was reduced in the dusty region in both the convective and stratiform regimes. At temperatures below 0°C, convective reflectivity was reduced in the dusty regions while stratiform reflectivity was increased. Min et al. (2009) performed an additional sensitivity test to differentiate the effects of dynamics on hydrometeor growth and precipitation formation from the microphysical effects of dust. The sensitivity test revealed that, in the absence of dust, relatively stronger convective intensity also resulted in higher stratiform reflectivity values. This indicated that the reduced reflectivity in the convective regime and increased stratiform reflectivity observed in the dust sector were a result of changes to microphysical processes rather than dynamics. These microphysical changes were suggested to be a result of increased heterogeneous ice formation, which delayed the formation of large precipitation particles in the convective regime until sufficient growth occurred during transport into the stratiform regime to pass the minimum reflectivity threshold (Min et al., 2009).

The use of bin microphysics in our WRF-SBM model allows us to simulate radar reflectivity directly from each respective hydrometeor's size distribution. These PSDs evolve naturally within the model during cloud formation, growth/evaporation of particles, conversion of cloud mass into precipitation, and eventual removal of precipitation particles by sedimentation.





Therefore, a more accurate depiction of microphysical changes to precipitation formation and the associated changes to radar reflectivity is possible in comparison with the fixed hydrometeor PSDs used in bulk radar simulators. To compare the observations of Min et al. (2009) with our results, we have recreated similar CFAD plots using model derived reflectivity. Figure 5 describes the radar reflectivity CFADs of the convective and stratiform regimes for the Clean and three dust cases.

As IN concentration is increased in the simulations, changes in ice formation and growth processes result in decreased convective reflectivity at temperatures above 0°C. Likewise, stratiform reflectivity at temperatures below 0°C is increased in the dust cases. These changes suggest that increased heterogeneous ice formation is significantly affecting the formation of precipitation sized particles consistent with the hypothesis of Min et al. (2009). We note that convective reflectivity at temperatures below 0°C and stratiform reflectivity at temperatures above 0°C are both increased in the dust cases compared

to the Clean case. This differs from the reduced reflectivity values reported in Min et al. (2009) and Li and Min (2010) for these locations. These differences can be partially explained by greater water vapour content within the dust layer in the model simulations compared to the observed SAL.

Measurements from AIRS/AMSU/HSB indicate that the relative humidity in the dust layer is about 20% drier than the

surrounding air. While a dry air layer is present in the WRF's initial and boundary conditions, the model slightly overestimates precipitable water compared to observations (Figure 1). To examine the impacts of dust layer moisture content on our case study, we have conducted additional numerical simulations based on the D1.2 case. The dust layers within these test cases feature relative humidity values that are 5% drier than the original D1.2 case. The first case (Dry5init) reduces the water vapour content in the dust layer over the entire 4th domain at model start-up time. The boundary conditions entering

the 4th domain are unchanged from the original D1.2 case. The second case (Dry5bound, not shown) reduces water vapour content at the boundaries of the 4th domain for the duration of the simulation with no changes made to the dust layer's initialized moisture content at model start-up time. Figure 5 describes the convective and stratiform CFADs of the D1.2 and Dry5init cases. The first and second columns describe the D1.2 CFAD and the D1.2 minus Clean case difference plots, respectively. The third column describes the Dry5init minus Clean case difference plots. Reduced moisture content in the

Dry5init case weakens convective cloud formation, which decreases convective reflectivity overall at temperatures below 0°C and shifts reflectivity to lower values at temperatures above 0°C. Reflectivity in the stratiform regime is still increased compared to the Clean case at temperatures below 0°C, but is also shifted to lower values at temperatures above freezing. These changes are very similar to the observed changes of convective and stratiform reflectivity described by Min et al. (2009) and Li and Min (2010). The Dry5bound case results in similar changes as those described by the Dry5init case,

although with greater reductions in the convective regime and smaller increases in the stratiform regime as a result of the drier boundary air transported into the 4th domain for the duration of the simulation.

### 4.3 Effects on Primary Ice formation and Hydrometeor Number Concentrations

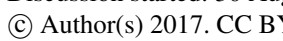



The convective core is the primary determiner of cloud macrophysical properties such as cloud top height and anvil cloud area (Futyan and Del Genio, 2007). However, changes to cloud microphysical processes resulting from AIE will modulate these macrophysical properties differently depending on the aerosol ice/liquid nucleation activity. In our numerical simulations, increasing IN in the dust cases increases the total number of new ice crystals forming in the heterogeneous

nucleation regime between -5°C and -38°C (Figure 8b to Figure 8d). This affects the vertical distribution of cloud ice particles by changing the locations of initial ice formation and subsequent growth. Figure 7a and Figure 7b describe the vertical distribution of ice particles formed by the model's heterogeneous and homogeneous ice formation schemes in the convective and stratiform cloud regimes, respectively. The ice formation number at each vertical level is summed horizontally and with respect to time for each cloudy pixel in the specified cloud regime and is represented by a $\log_{10}$ value.

Figure 7c describes the vertical distribution of residual (non-activated) IN number concentration in the convective cloud regime. This value is averaged over all convective cloud data points and temporally over the duration of the simulation. Increasing IN concentration in the convective core results in significant increases in ice formation between -5°C and -15°C. Ice formation in this temperature range can deplete available IN (Figure 7c) and reduce heterogeneous ice formation between -15°C and -38°C. This depletion effect is substantial between ~7km and 11km in the D.12 and D1.2 cases. When the IN

concentration is sufficiently high, such as in the D12 case, depletion is not as significant as in the other cases and ice formation is significantly increased over the majority of the -5°C to -38°C temperature range. At the -38°C threshold, ice formation number is progressively reduced as IN number is increased, which suggests that clouds are glaciating at warmer temperatures compared to the Clean case. The percentage of ice formed by homogeneous freezing out of total ice formation in each of the four cases is: Clean (91.32%); D.12 (91.24%); D1.2 (47.86%); D12 (0.02%). The reduction of homogeneous

freezing is due both to fewer liquid drops crossing the -38°C threshold (Figure 8j to Figure 8l) and reduced peak supersaturations resulting from increased ice growth at temperatures above -38°C. Finally, we note that stratiform ice formation is also increased in the dust cases compared to the Clean case. The increase, while not as large as in the convective core, contributes to increased cloudiness in the stratiform regime between -5°C and -38°C by increasing local concentrations of small, slow-falling ice crystals (Figure 12).

Changes to the location and number concentration of initial ice particle formation affect the vertical distribution of ice and liquid hydrometeors in several ways. Figure 8a to Figure 8d describes the time evolution of convective averaged ice and snow particle number concentration. Increasing IN concentration results in a greater number of ice/snow particles in the heterogeneous nucleation regime and a corresponding reduction within the cloud column at temperatures below -38°C. This

indicates that the reduced homogeneous ice formation number noted in Figure 7a is not counteracted by the transport of a similar number of particles from temperatures warmer than -38°C. While more particles are formed in the heterogeneous nucleation regime between -5°C and -38°C compared to the Clean case, there are also more opportunities for these particles to collide and be incorporated into larger particles. For example, more frequent riming of ice and snow particles in the dust cases increases the formation of graupel (Figure 8e – Figure 8h). More frequent riming in turn reduces the average number



of liquid drops in the convective regime at temperatures colder than -5°C (Figure 8i – Figure 8l). While dust only activates as IN and not CCN in our simulations, average liquid drop number at temperatures above -5°C is affected by the more numerous ice particles forming in the heterogeneous nucleation regime and subsequently melting after falling into warmer temperatures. Small ice particles melt into small drops that may evaporate, while large drops formed from melted

snow/graupel may collect smaller drops by collision-coalescence or break up into smaller drops themselves.

## 4.4 Effects on Convective PSDs and Collection Processes

Increasing the total number of ice particles formed in the heterogeneous nucleation regime affects the PSD in two ways.

First, available water vapour is partitioned over a greater number of smaller particles. Second, these smaller particles are less efficient at colliding with other particles. Both effects reduce the growth rates of the individual particles and shift the PSD to smaller particle sizes overall. The SBM allows us to examine the effects of dust on the PSD of the different hydrometeors without creating an arbitrary distinction between cloud and precipitation sizes particles. Dust related changes to the bin PSD of each hydrometeor type are described in Figure 9. The provided radii values of the represented hydrometeor species are

derived from the pre-calculated bin radii values used by the model, which are based on assumed particle densities and the mass doubling relationship between the individual bins. Contour values represent $\log_{10}$ values of bin number concentration. The difference plots likewise describe the relative change of these $\log_{10}$ values, representing $\mathrm{Log}_{10}$ (Dust/Clean) values. As dust in our study acts as IN exclusively and not CCN, we focus our discussion on the -5°C to -38°C degree range conducive to heterogeneous nucleation and freezing. Since dust in nature can also act as effective CCN and may therefore be removed

from the system by warm rain processes before freezing occurs, these results should be interpreted as an upper range of IN effects for a given dust number concentration.

Figure 9a, Figure 9e, and Figure 9i describe time series of the PSD averaged over convective data within the -5°C to -38°C temperature range for ice/snow, cloud/rain drops, and graupel. The remaining plots in Figure 9 describe the differences

between the three dust cases and the Clean case. The addition of IN to the DCC system produces an initial burst of ice formation covering the range of the PSD. In the D.12 and D1.2 cases, this is followed by a reduction in the small crystals and an increase in larger crystals and snow between hours 12 and 24. IN concentration has been depleted during this time period, which reduces the formation of small ice crystals. Existing ice crystals grow by particle collection into snow, hence the upwards slope in the difference contours between hour 12 and 24. When IN concentration is sufficiently large (D12 case)

depletion is not as significant (Figure 7) and small ice crystals continue to form over the duration of the simulation. The liquid PSD describes an enhancement to the largest drop sizes that could be the result of increased collision-coalescence of available drops and/or the recirculation of recently melted large ice particles to temperatures below freezing. Stronger vertical motion in the dust cases may also transport more large drops from temperatures above 0°C directly. The middle size range of the liquid PSD is reduced though the duration of the simulation, corresponding with the enhanced bin population in





the graupel PSD. The formation of graupel in our model occurs by two distinct mechanisms: direct freezing of large liquid drops, by the homogeneous or immersion/contact freezing mechanisms, and collisions between liquid and ice particles. There is evidence of increased large drop freezing, as seen by the enhancement to the largest bin sizes in the graupel PSD. However, the majority of graupel particles are formed by riming, as seen by the similar locations of reduction and

enhancement between the liquid and graupel PSDs. While riming is more frequent in the dust cases, as evidenced by the increasing graupel number concentration, the graupel sizes shift smaller. This is a result of both overall smaller ice crystals sizes and competition between the individual particles for available liquid drops during riming, reducing growth rates (Figure 10).

Particle collection processes are the primary source of precipitation formation due to the more rapid accumulation of mass compared to purely diffusional growth. In liquid clouds, collision-coalescence processes allow cloud drops to collect into rain drops. In ice and mixed phase clouds, ice-ice (aggregation) collisions and ice-liquid (riming) collisions become more frequent as total frozen particle number concentration increases. Figure 10 describes the changes to aggregation (row 1), riming (row 2), and drop autoconversion (row 3) in the convective regime with respect to time for the Clean and dust cases.

Drop autoconversion rate (1.e-4 $kg^{-1}$ $s^{-1}$) tracks the formation of rain drops from cloud drops by collision-coalescence processes. Aggregate number ($kg^{-1}$) tracks the change of ice particles before and after aggregation occurs and is more negative for a more efficient process. Riming rate (g $kg^{-1}$ $s^{-1}$) tracks the liquid mass converted to graupel through the riming process and, again, is more negative for a more efficient process. These two processes are also affected by the relative availability of liquid and ice content within the cloud. As riming can only occur where ice and liquid particles coexist, this

limits the most significant riming to the convective core below the cloud's glaciation level. Likewise the drop collision-coalescence processes are reduced in the heterogeneous nucleation regime in the dust cases due to the conversion of liquid content into ice at temperatures below 0°C. In the stratiform regime, relatively little liquid content is transported from the convective core due to the majority of freezing occurring in the core itself. Therefore, ice-ice particle interactions are the most common in the stratiform regime and snow is the predominant precipitation particle type (Stith et al., 2002; Heymsfield

et al., 2002; Lawson et al., 2010, Gallagher et al., 2012).

In the Clean case the majority of ice forms by homogeneous freezing, which limits significant ice-ice particle interactions in the heterogeneous nucleation regime until a significant number of ice particles have fallen down from the homogeneous freezing regime. The small addition of IN in the D.12 case forms a sufficient number of ice particles to increase aggregation

activity before hour 18 near the 0°C freezing level, but a noticeable gap at higher altitudes in the heterogeneous nucleation regime remains due to more significant homogeneous freezing compared to the other dust cases. Increasing the IN concentration further results in maximum values near 0°C and decreasing upwards to colder temperatures. The significantly larger values of aggregation number in the D12 case compared to the other cases (Figure 10a to Figure 10d) is a result of the greater number of ice crystals forming at warmer temperatures where particle "sticking" efficiency is higher (Hallgren and



Hosler, 1960). While aggregation is the primary precipitation process in the stratiform regime, the aggregation numbers in this regime are smaller than in the convective regime. This is a result of the significantly greater number of ice crystals that form initially in the core and are subsequently collected into snow particles before being transported into the stratiform regime.

The effect of increased heterogeneous ice formation on the efficiency of riming is tied into both the size and number of ice particles that form and the overall availability of liquid water drops. The larger midlevel liquid water content in the Clean case results in efficient riming despite the lower ice number in the heterogeneous nucleation regime compared to the dust cases. Increased ice formation in the heterogeneous nucleation regime increases riming rates near the 0°C freezing level.

This is due to the greater total number of ice particles and the significant presence of liquid water content near the melting level. Above 6km in the convective regime, where ice formation becomes significant in the dust cases, riming rates become progressively lower as IN concentration is increased. The smaller sizes of ice particles forming in these locations reduce the collision efficiency between ice particles and liquid drops. The reduced number of available liquid drops in the dust cases also affects riming rates by decreasing the depth of the mixed-phase environment in which riming may occur. Changes to

drop autoconversion rates are similarly affected by changes to drop number concentration and PSD. While the current case studies do not allow for dust particles to activate as both CCN and IN, changes to collision-coalescence processes result from changes to ice formation and the subsequent impact on liquid water mass both above and below the freezing level. In general the addition of IN to the dust cases results in lower liquid water content in the heterogeneous nucleation regime due to riming and immersion/contact drop freezing which limits the opportunities for collision-coalescence to occur. At altitudes

below 6km, collision-coalescence rates are affected by the number and PSD of ice particles that melt after falling into above freezing temperatures. We note that higher autoconversion numbers occur at temperatures slightly above 0°C between hour 15 and 20 in the D1.2 (row 3c) and D12 (row 3d) cases. These increases are also visible in the changes to vertical rain rates at these temperatures (Figure 14).

### 4.5 Changes to Convective Intensity and Core Top Height

The formation of smaller and more numerous cloud ice particles in the heterogeneous nucleation regime results in increased latent heat release in the convective core between -5°C and -38°C. This is due to both the diffusional growth of frozen

particles and latent heat released by the phase change occurring during riming. Diffusional growth is the source of the majority of latent heat release and may consume much of the updraft's available water vapour. Increased latent heat release invigorates convective updrafts compared to the clean case. Figure 11a and Figure 11d describe the time evolution of convective regime averaged updraft and downdraft velocity. Figure 11b and Figure 11e (Figure 11c and Figure 11f) describe the average latent heat (water vapour mixing ratio) at temperatures < 0°C within the updrafts and downdrafts, respectively.



As IN concentration is increased, average convective updraft intensity is progressively increased between hour 10 and 20. Likewise, updraft latent heat is increased and updraft water vapour content is reduced. This is consistent with increased diffusional growth of the more numerous particles that form in the dust cases. Increased convective updraft velocity in the dust cases results in higher convective core top heights from model hour 6 to about model hour 20. During the transition to

deep convection between hour 6 and hour 12 the core top height increase is fairly linear for increasing IN concentrations. The time averaged convective core height (cloud tops < 0°C) between hour 6 and 12 are: Clean (8.91 km); D.12 (8.93 km; +0.25%); D1.2 (9.28 km; +4.2%); D12 (9.34 km; +4.8%). Despite the invigorated updrafts occurring throughout the hour 6 to hour 20 time period, the core cloud top height is also affected by changes to the ice/snow PSD between hour 12 and hour 20 (Figure 9). The average convective core height (cloud tops <0°C) between hour 12 and 20 are: Clean (12.1 km); D.12

(12.25 km; +1.2%); D1.2 (12.04 km; -0.5%); D12 (12.61 km; +4.2%). Note that the average core height in the D1.2 case is lower than the Clean case during this time period due to the presence of more large and fewer small sized particles (Figure 9c) as a result of the IN depletion described in Figure 5. This limits the number of particles that remain aloft in the D1.2 case, due to faster sedimentation rates of the large particles. Stronger downdrafts occurring between hour 10 and 20 also increase evaporation/sublimation of the more numerous particles in the dust cases. This consumes latent heat and increases

water vapour content within the convective downdrafts (Figure 11e; Figure 11f).

To summarize, increasing IN concentration in the dust cases results in increased ice formation and growth within the heterogeneous nucleation regime between -5°C and -38°C. Partitioning of available water vapour over more numerous particles shifts the PSD of cloud ice crystals to smaller sizes, which grow more slowly. The diffusional growth of these

particles increases latent heat release in the heterogeneous nucleation regime and invigorates convective updrafts. Homogeneous ice formation is reduced due to fewer liquid drops crossing the -38°C threshold as well as reduced peak supersaturation due to ice growth within the heterogeneous regime. Despite reduced homogeneous ice formation, invigorated updrafts result in higher convective core cloud tops overall compared to the Clean case.

**4.6 Effects on Stratiform Cloud Regime**

The macrophysical and microphysical properties of the stratiform and anvil cloud regime are significantly affected by cloud and precipitation formation processes initiated within the convective core and are also affected by changes to local ice formation within the stratiform/anvil regime itself. Invigorated updrafts in the dust cases carry a greater number of both large

and small particles in the convective core to the level of divergence. These particles are then transported by wind shear into the milder updrafts of the stratiform regime. The large particles quickly sediment out and the smaller particles remain aloft. Figure 12 describes the stratiform ice/snow bin distribution as Figure 9 described the convective ice/snow bin distribution. Between hour 6 and hour 12 in the dust cases, the initial burst of ice formed by heterogeneous nucleation in the core is transported into the stratiform regime in conjunction with local ice formation. This results in increased bin populations over



much of the ice/snow PSD. After hour 12 until about hour 26, the formation of small ice particles is reduced due to the depletion of IN by ice formation earlier in the simulation. Snow particles formed in the convective core grow to larger sizes during transport into the stratiform regime. This increases the relative bin populations at sizes between 1900um and 20000um compared to the convective regime. These large particles efficiently capture other smaller particles, resulting in the

greater reduction of smaller sized particles in the stratiform PSD compared to the convective regime (Figure 9c; Figure 12c). When IN concentrations are not as significantly depleted, such as in the D12 case, heterogeneous nucleation produces additional small ice crystals throughout the hour 12 to hour 26 period (Figure 12d). While larger sized particles continue to form in the D12 case, the location of the most significant enhancement to bin population shifts to smaller particle sizes compared to the D1.2 case due to competition between the more numerous particles during collection processes.

Many hydrometeors in the stratiform and anvil cloud regime were initially formed in the convective core and were transported into the stratiform/anvil regime by wind shear. However, increasing IN concentration in the dust cases also results in increased heterogeneous ice formation within the stratiform/anvil regime itself (Figure 7b) which affects cloudiness. Figure 13 describes the cloud occurrence numbers for the convective, stratiform and non-precipitating anvil

cloud regimes. Cloudiness is determined by the sum of all condensate mixing ratios within a grid box exceeding $10^{-6}$ kg/kg. The vertical distribution of convective cloud occurrence increases between -5°C and -38°C as IN concentration is increased in the dust cases. Likewise, stratiform cloud occurrence is increased between -5°C and -38°C due to both increased transport from the convective core and increased heterogeneous ice formation within the stratiform regime itself. However the anvil cloud is significantly affected by changes in hydrometeor PSDs. A small IN concentration in the dust layer (D.12) results in

greater anvil cloud occurrence compared to the Clean case. In the D.12 case, due to the limited supply of IN, the formation of large ice particles is not significantly increased compared to the small ice particles that form. The small ice particles are transported greater distances in the updrafts and sediment out slowly, which results in a higher (Figure 4) and broader cloud distribution compared to the Clean case. In the D1.2 case, some ice particles are transported from the core and more ice particles are formed locally through heterogeneous formation processes with available IN in the stratiform regime. Some of

the particles grow by collection processes to large sizes (Figure 12c). These large particles sediment out quickly in the weaker updrafts of the stratiform regime and therefore are not transported into the anvil cloud regime. In conjunction with reduced homogeneous ice formation, this results in fewer particles forming locally within and/or being transported into the anvil regime. Therefore the stratiform/anvil cloud top distribution in the D1.2 case is lower and narrower compared to the Clean case and other dust cases. The D12 case is affected by both the formation of more numerous ice large particles

(compared to the D.12 case) and more numerous small ice particles (compared to the D1.2 case). The strong updraft intensities in the D12 case transport significant condensate mass into the stratiform regime. The large particles that form in the D12 case sediment out quickly, but the small ice particles remain near the cloud tops. This results in a stratiform/anvil cloud top distribution that is lower and less broad compared to the Clean and D.12 cases, but is higher and wider than the D1.2 case. However, after hour 20 (Figure 12d) the ice particles in the D12 case grow to large sizes and sediment out. This



results in the lower stratiform/anvil cloud top height from hour 20 until the end of the simulation compared in the D12 case to the D1.2 case (Figure 4).

## 4.7 Vertical Rain Rates and Surface Accumulation

Increasing heterogeneous ice formation by increasing IN concentrations results in larger ice mass near the 0°C temperature level, but greater competition between individual particles for water vapour and available small drops/crystals for collection shifts the formation of precipitation sized particles to higher altitudes. Smaller particles that sediment out or are transported below the melting level are more likely to evaporate below the cloud due to a slower fall speed. These changes result in a

reduced surface accumulation and enhanced rain rates above the freezing level. Figure 14 describes the accumulated surface rain rates (Figure 14a, Figure 14b) and rain rate vertical profile differences (Figure 14d, Figure 14e) for convective (column 1) and stratiform (column 2) regimes. Figure 14c describes the time series of total accumulated surface precipitation, while Figure 14f describes the total fraction of precipitation formed at each vertical level, for the Clean and dust cases. In general, the addition of IN reduces the average surface rain accumulation for the convective (Figure 14d) rain regime and increases it

for the stratiform (Figure 14e) rain regime. This is due to the different effects of dust on the primary sources of precipitation in the two regimes. The percent reduction of total surface precipitation in the dust cases from the Clean case values at the end of the simulation are: D.12 (-1.14%); D1.2 (-3.95%); D12 (-6.02%).

Convective rain is significantly affected by changes to graupel formation, which in the dust cases is shifted towards smaller

sizes (Figure 9). The smaller graupel sizes is a result of decreased riming rates above 6km due to smaller ice particle sizes and lower liquid water content (Figure 8). By the end of the simulation, convective surface precipitation accumulation is reduced from the Clean case as follows: D.12 (-2.3%); D1.2 (-5.5%); D12 (-7.9%). In the stratiform regime precipitation is predominantly a result of snow formation. In the dust cases, snow formation is enhanced due to the increased transport of ice mass from the convective core and the warmer glaciation temperatures in the convective regime. This initiates the

aggregation processes earlier in the simulation and at warmer temperatures than in the Clean case (Figure 10). Stratiform surface precipitation accumulation is increased from the Clean case value as follows: D.12 (+10.1%); D1.2 (+8.2%); D12 (+13.1%). At altitudes below 6km, collision-coalescence rates are affected by the number and PSD of the frozen particles that melt in the above-freezing temperatures. In the convective regime, increased aggregation rates (Figure 10) and freezing of large drops to graupel (Figure 9) result in higher autoconversion rates in the D12 case compared to the other dust cases

between ~1km and the 0°C freezing level when these large particles melt. This partially counteracts the reduced rain rates between 4 and 8 km resulting in near surface rain rates that slightly exceed the D1.2 case, although final surface accumulation is still lower in the D12 case due to the greater reductions at higher altitudes.



# 5 Conclusions

The MCS occurring on 08 March 2004 in the tropical eastern Atlantic, first described in Min et al. (2009) was simulated using the WRF model with a spectral-bin microphysical scheme. Ice nucleation parameters within the SBM were updated to connect heterogeneous and homogeneous ice formation with IN to investigate the effects of dust acting as IN. In the first of a
two part study, we present the effects of IN activation on ice formation processes and the eventual effects on the large-scale cloud fields. The hypothesis of Min et al. (2009) suggested that dust particles forming ice at heterogeneous temperature ranges (-5°C to -38°C) results in changes to precipitation formation processes and ice particle size distributions shifting to smaller sizes in the heterogeneous nucleation regime. Lower stratiform/anvil cloud top heights were reported (Min and Li, 2010), despite the presence of more numerous deep clouds with large IWP which suggests that convective invigoration
(increased latent heat release; stronger updrafts) is occurring.

Increasing IN concentration in the dust case simulations results in the formation of a greater number of ice particles in the convective core between -5°C and -38°C compared to the Clean case (Figure 8). The partitioning of available water vapour over the greater number of particles results in smaller ice crystal and graupel sizes in the dust cases (Figure 9). The ice
particles grow more slowly due to the increased competition between individual particles for available water vapour (Figure 11). Latent heat release in the heterogeneous nucleation regime is increased in the dust cases due to diffusional growth and liquid-to-ice phase changes during riming of the smaller, more numerous particles. Convective updrafts are invigorated (Figure 2; Figure 11), resulting in increased overshooting and higher convective core top heights. The increased downdraft velocity and more numerous small particles result in increased evaporation/sublimation and latent cooling (Figure 11).

Particle growth resulting from collection processes is also reduced, due to the lower collision efficiency of the smaller particle sizes in the dust cases. Therefore precipitation formation is shifted to colder temperatures (higher altitudes) within the heterogeneous nucleation regime. When available IN concentration in the dust cases is depleted, the formation of new ice crystals in the heterogeneous nucleation regime is limited. Collection processes remove small ice crystals formed earlier in
the simulation and increase the formation of large ice/snow particles in both the convective (Figure 9) and stratiform (Figure 12) regimes. This is most visible in the D1.2 case between hour 12 and hour 18. When few small ice particles remain aloft, due to reduced homogeneous ice formation and/or increased particle collection, stratiform/anvil cloud top heights will be lower over the majority of the simulation, as in the D1.2 and D12 cases (Figure 4; Figure 13). When small particles are relatively more numerous and the number of large particles is not significantly affected, such as in the D.12 case,
stratiform/anvil cloud top heights are higher than in the Clean case (Figure 4; Figure 13). The small particles in the D.12 case are transported to higher altitudes in the convective updrafts and remain aloft for longer times. More numerous but smaller graupel particles form in the dust cases (Figure 3; Figure 9) due to the reduced riming efficiency of small ice particles (Figure 10) and increased competition between the individual frozen particles during riming for available liquid drops. The



greater heterogeneous ice numbers also increase ice particle aggregation in the -5°C to -38°C temperature range (Figure 10), leading to increased snow formation in both the convective and stratiform regimes (Figure 9; Figure 12). Growth competition between the more numerous individual particles during riming/aggregation shifts precipitation formation to higher altitudes within the heterogeneous nucleation regime. This results in changes to simulated reflectivity values (Figure 2; Figure 5) which are similar to observed effects on reflectivity (Min et al., 2009; Li and Min, 2010).

The impacts of dust as IN on model simulated reflectivity are mostly consistent with observed changes, i.e., dust cases producing smaller reflectivity values near the surface and larger values above the freezing level and most significantly in the stratiform regime (Figure 2; Figure 5). Radar reflectivity in the dust cases is affected by PSDs shifted to smaller sizes, reduced particle fall rates, and increased formation of large snow particles. The contribution of graupel and rain drops to total reflectivity in the dust cases is reduced due to the shift to smaller particle sizes (Figure 3; Figure 9) and reduced drop concentrations (Figure 8), respectively. This decreases dust case reflectivity values at temperatures above 0°C in the convective regime (Figure 2, Figure 5). Snow particles have large radii compared to graupel and rain drops of comparable mass (Figure 3; Figure 9) and have slower fall rates. More numerous large snow particles in the dust cases result in increased reflectivity values at temperatures below 0°C (Figure 5), most notably in the stratiform regime where aggregation is the dominant precipitation formation process. The dust case reflectivity CFADs differed from observed reflectivity changes in the convective regime (>0°C) and in the stratiform regime (>0°C). Specifically, reflectivity in these locations is increased in the dust case simulations while observations indicate that reflectivity is reduced. Higher moisture content in the dust layer compared to the observed test cases was suggested as a possible cause of these differences. Additional test cases based on the D1.2 case were simulated to determine the effects of reduced moisture content within the dust layer on model results. Reducing dust layer moisture content by 5% (Dry5init case) was sufficient to weaken convective cloud formation and affect the resulting reflectivity CFADs (Figure 6) in ways consistent with observed changes (Min et al., 2009; Li and Min, 2010). Convective reflectivity (<0°C) and stratiform reflectivity (>0°C) were both reduced compared to the Clean case. Stratiform reflectivity at temperatures below 0°C was also increased from the Clean case, indicating that microphysical changes to cloud and precipitation formation processes are similar to those in the original D1.2 case.
.

### Acknowledgements

This work was supported by the NSF under contract AGS-1138495 and PIRE, by US DOE's Atmospheric System Research program (Office of Science, OBER) under contract DE-FG02-03ER63531, and by the NOAA Educational Partnership Program with Minority Serving Institutions (EPP/MSI) under cooperative agreements NA17AE1625 and NA17AE1623. J. Fan is supported by the U.S. Department of Energy (DOE) Atmospheric System Research (ASR) Program. The Pacific



Northwest National Laboratory (PNNL) is operated for the DOE by Battelle Memorial Institute under contract DE-AC06-76RLO1830.

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

**Tables**

**Table 1:** WRF model parameterizations selected for use in study simulations.

| Selected Weather Research and Forecasting (WRF) model parameterizations | |
|---|---|
| **Parameterization** | **Selected option** |
| Microphysics | Domain 1,2,3: Thompson (Thompson et al., 2008); Domain 4: SBM (cited in text) |
| Cumulus | Domain 1,2: Kain-Fritsch (Kain, 2004) |
| LW Radiation | Rapid Radiative Transfer Model (Mlawer et al., 1997) |
| SW Radiation | Dudhia scheme (Dudhia, 1989) |
| PBL | MYNN2(Nakanishi and Niino, 2006) |
| Surface layer | MM5 similarity (Zhang and Anthes, 1982) |
| Land surface | RUC LSM (Smirnova et al., 1997) |





**Table 2:** Ship observed aerosol number concentrations from the AEROSE campaign corresponding to the March 2004 Saharan dust outbreak.

| March 2004 Ship Observed Aerosol Number | | | | | | |
|---|---|---|---|---|---|---|
| **Radius** | 0.3-0.5 | 0.5-1 | 1-5 | 5-10 | 10-25 | 25 | micron |
| **Dust-free** | 108.6 | 10.5 | 2.36 | 0.1029 | 3.00E-4 | 5.50E-8 | cm$^{-3}$ |
| **Dust** | 87.32 | 34.7 | 7.557 | 0.3537 | 1.45E-3 | 7.41E-6 | cm$^{-3}$ |

## 5  Figures

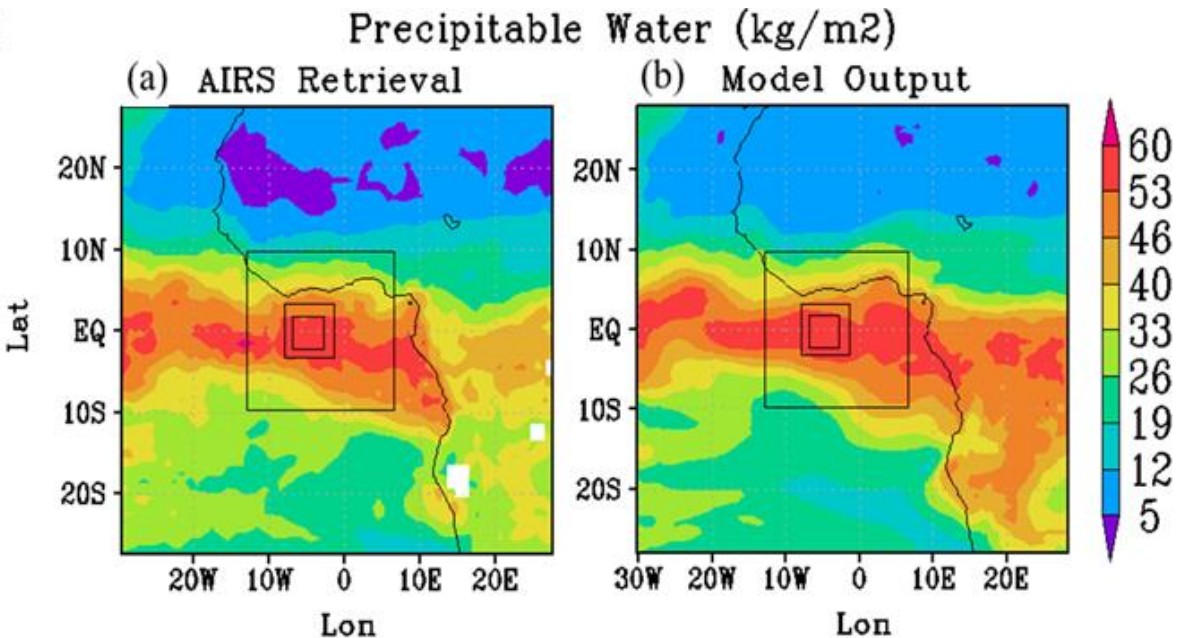

**Figure 1:** (a) AIRS total precipitable water averaged 07-09 March 2004, boxes denoting location of the three nested domains.

(b) Domain 1 model output precipitable water averaged 07-09 March 2004.



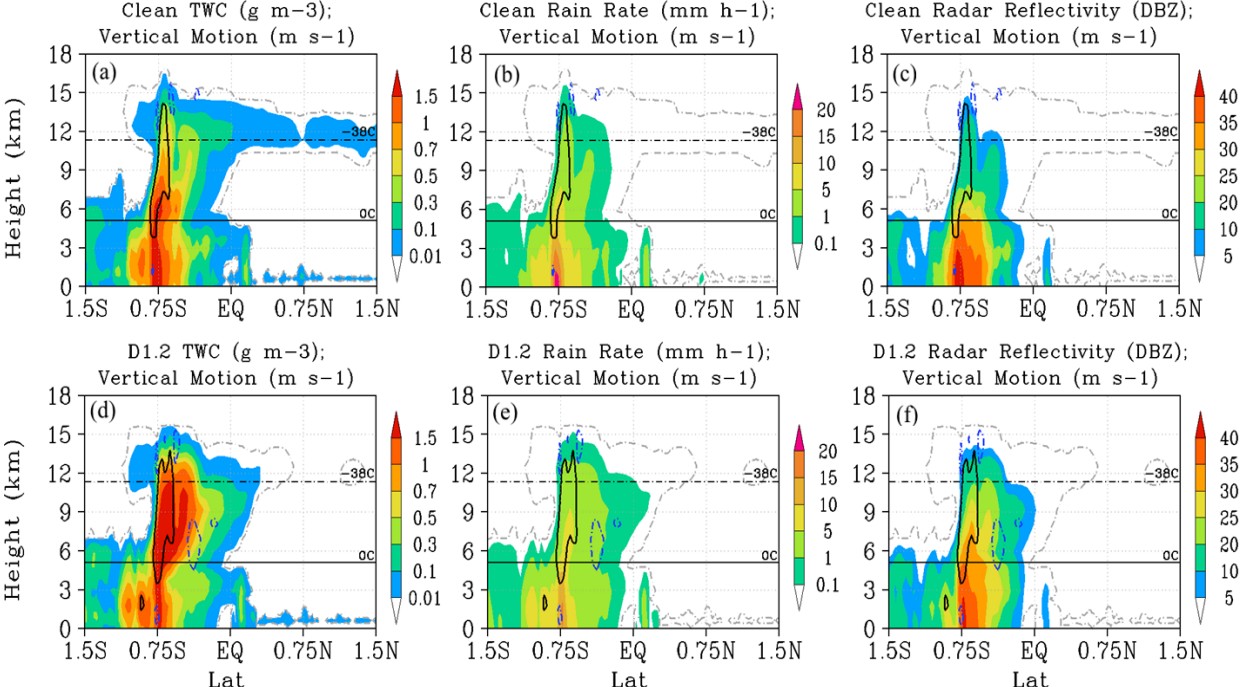

**Figure 2:** Zonally averaged longitude slice plot of similar DCC structures within the Clean (row 1) and D1.2 (row 2) cases. Shaded colours: total water content (TWC; column 1), vertical rain rate (column 2), and radar reflectivity (column 3); Line contours, all columns: vertical motion (solid black >1m/s; dashed blue <-0.1m/s); cloudiness threshold (dashed grey, >1e-6 kg/kg).



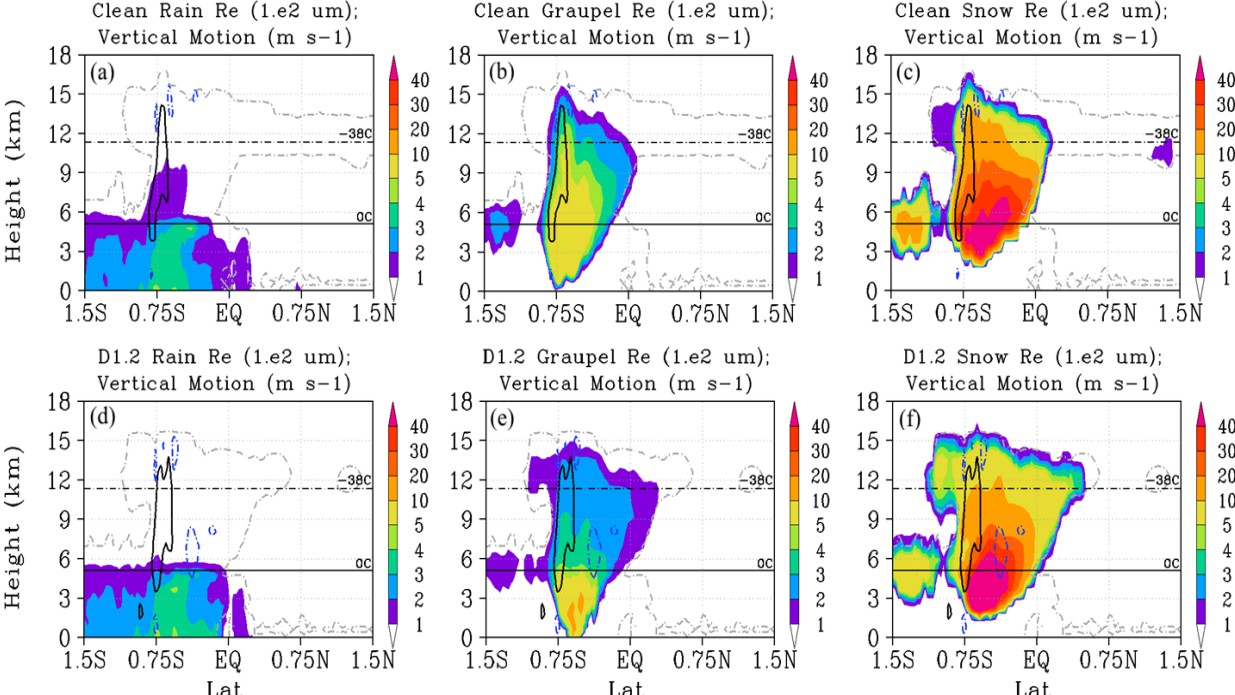

**Figure 3:** Slice plots representing same DCC as in Figure 1 for Clean (row 1) and D1.2 (row 2) cases. Shaded colours: rain drop effective radii (Re; column 1), graupel Re (column 2), and snow Re (column 3); Line contours, all columns: vertical motion (solid black >1m/s; dashed blue <-0.1m/s); cloudiness threshold (dashed grey, >1e-6 kg/kg).





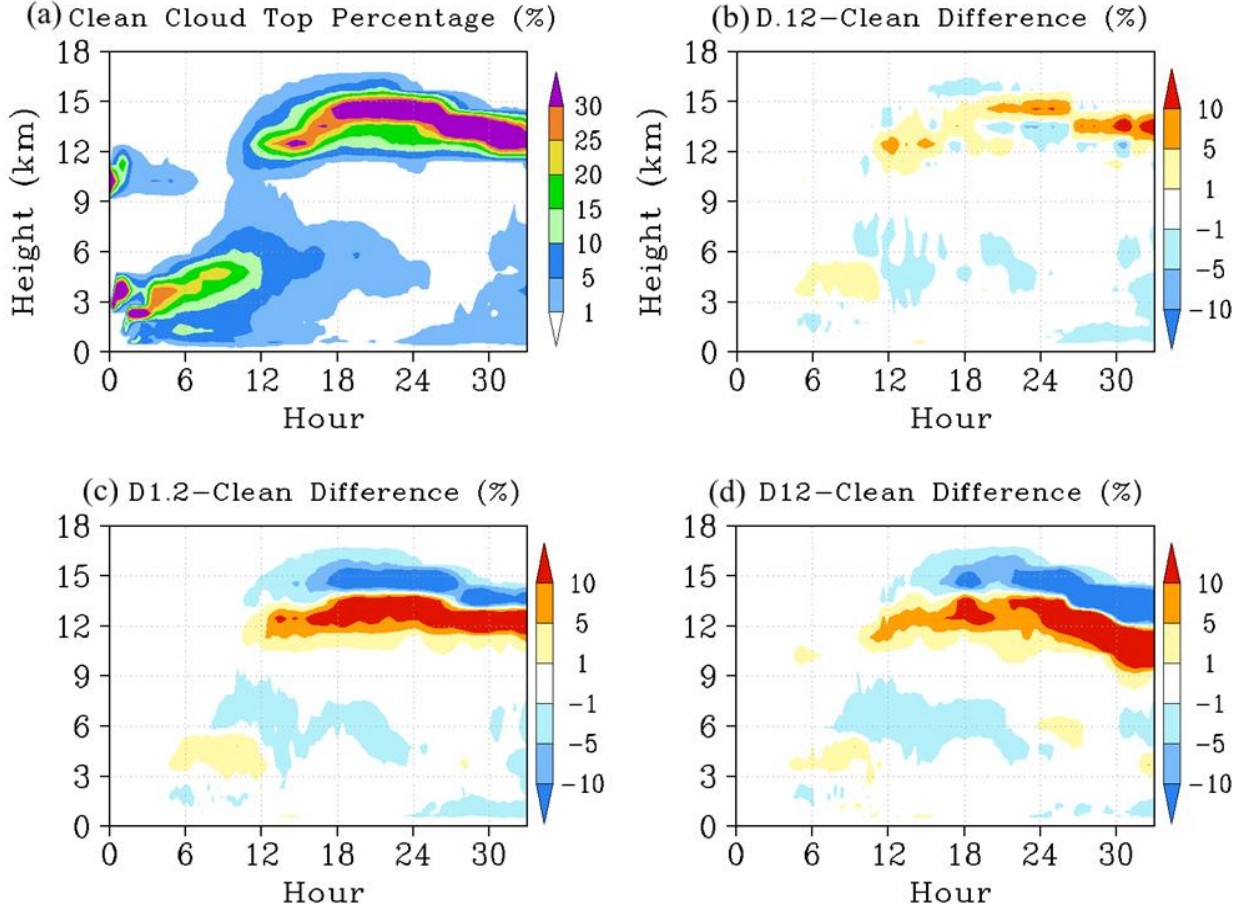

**Figure 4:** Time series of percentage of cloud tops occurring at each altitude for the (a) Clean case and the associated dust case minus Clean case differences plots for the (b) D.12 - Clean, (c) D1.2 - Clean, and (d) D12 - Clean cases.




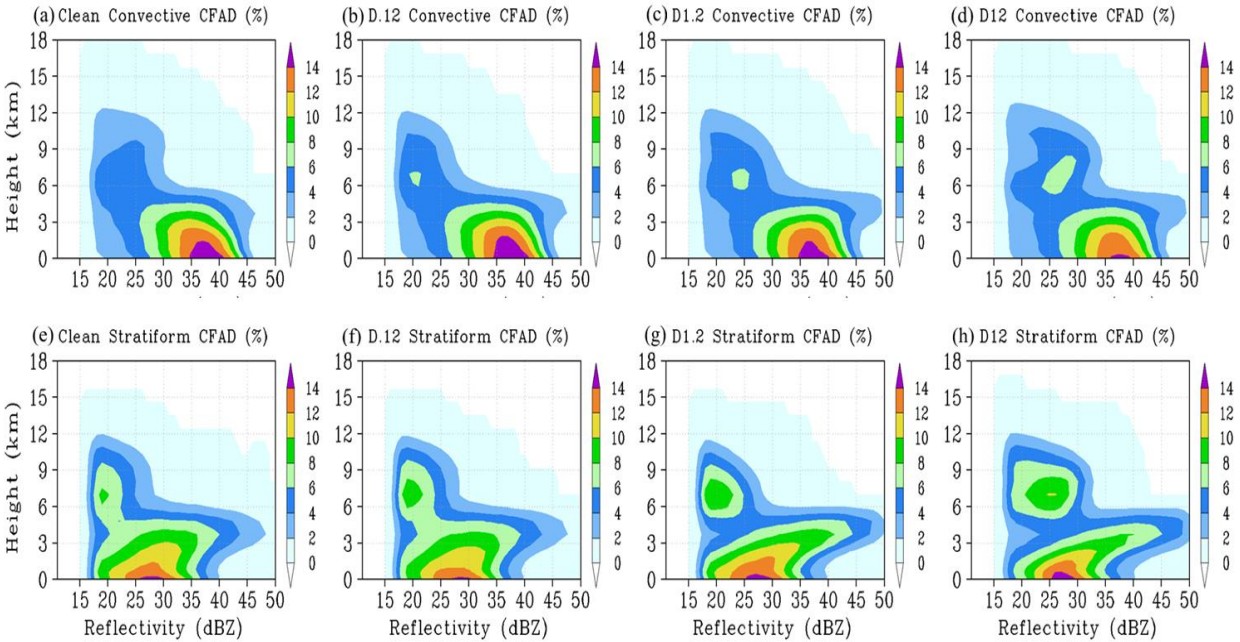

**Figure 5:** Contoured frequency by altitude diagrams (CFAD) of model simulated convective (tow row) and stratiform (bottom row) reflectivity. Columns: Clean, D.12-Clean, D1.2-Clean, D12-Clean cases, respectively.

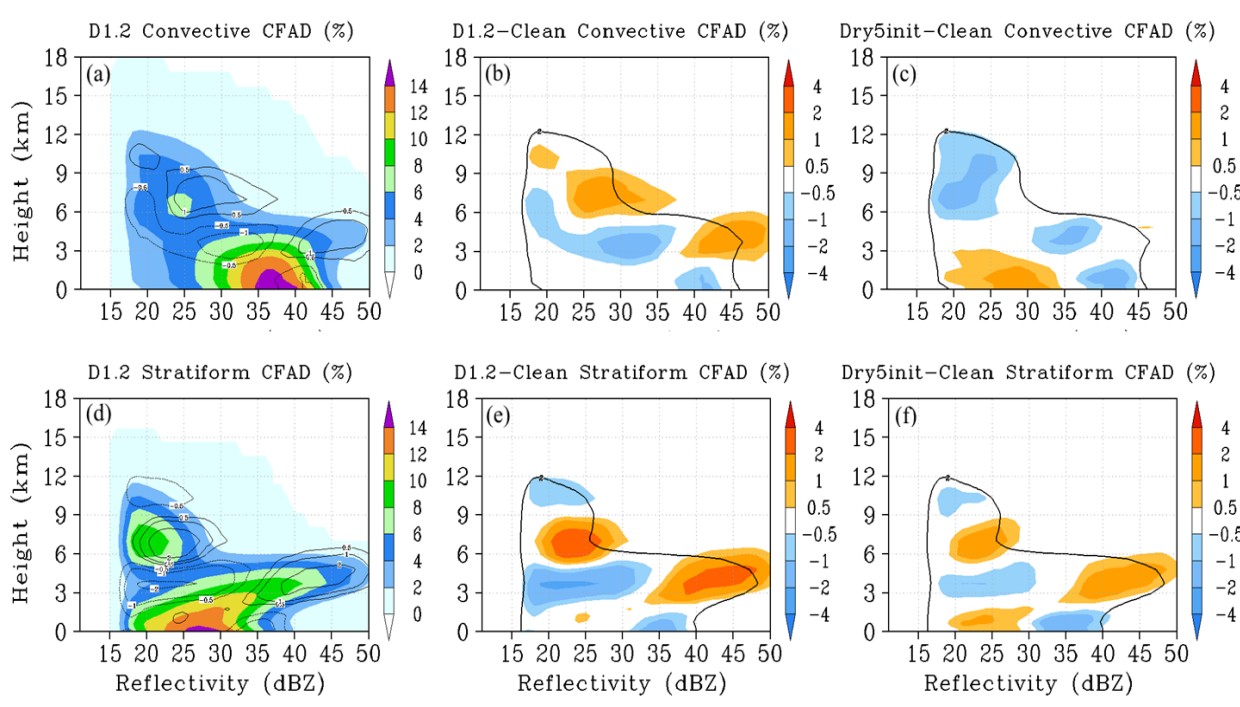





**Figure 6:** Contoured frequency by altitude diagrams (CFAD) of model simulated convective (tow row) and stratiform (bottom row) radar reflectivity. Columns: D1.2 case CFAD; D1.2-Clean case; Dry5init-Clean case. Dry5init case is based on the D1.2 case, but the dust layer is set to be 5% dryer at model start up. Boundary conditions are unchanged from the D1.2 case. Black contour line in difference plots represents the Clean case 2% contour value.

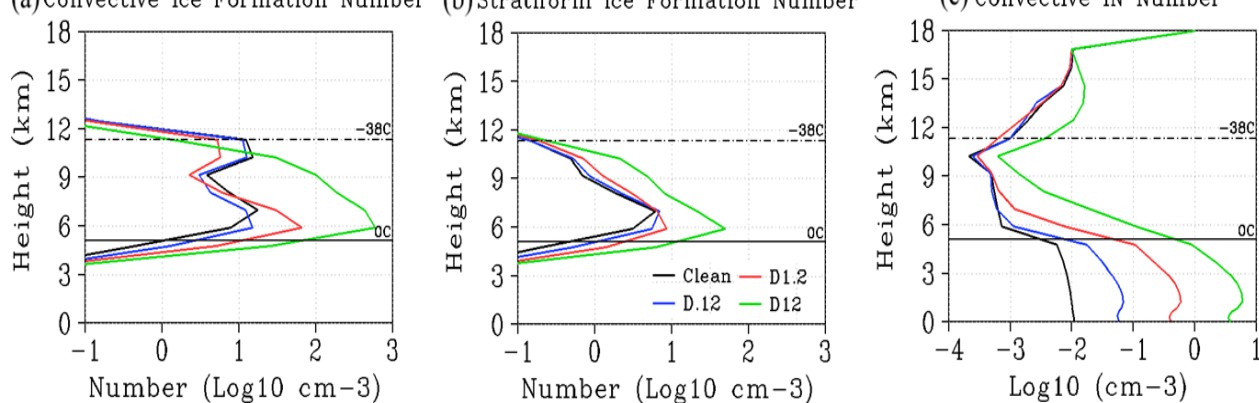

**Figure 7:** Vertical profile of combined heterogeneous and homogenous ice formation number (Log$_{10}$, cm-3) summed horizontally and over the duration of the simulation for convective (a) and stratiform (b) clouds, respectively. Convective cloud averaged and time averaged vertical profile of residual (non-activated, in-cloud) IN number concentration (Log$_{10}$, cm-3). Colours represent Clean (black), D.12 (blue), D1.2 (red), and D12 (green) cases, respectively.



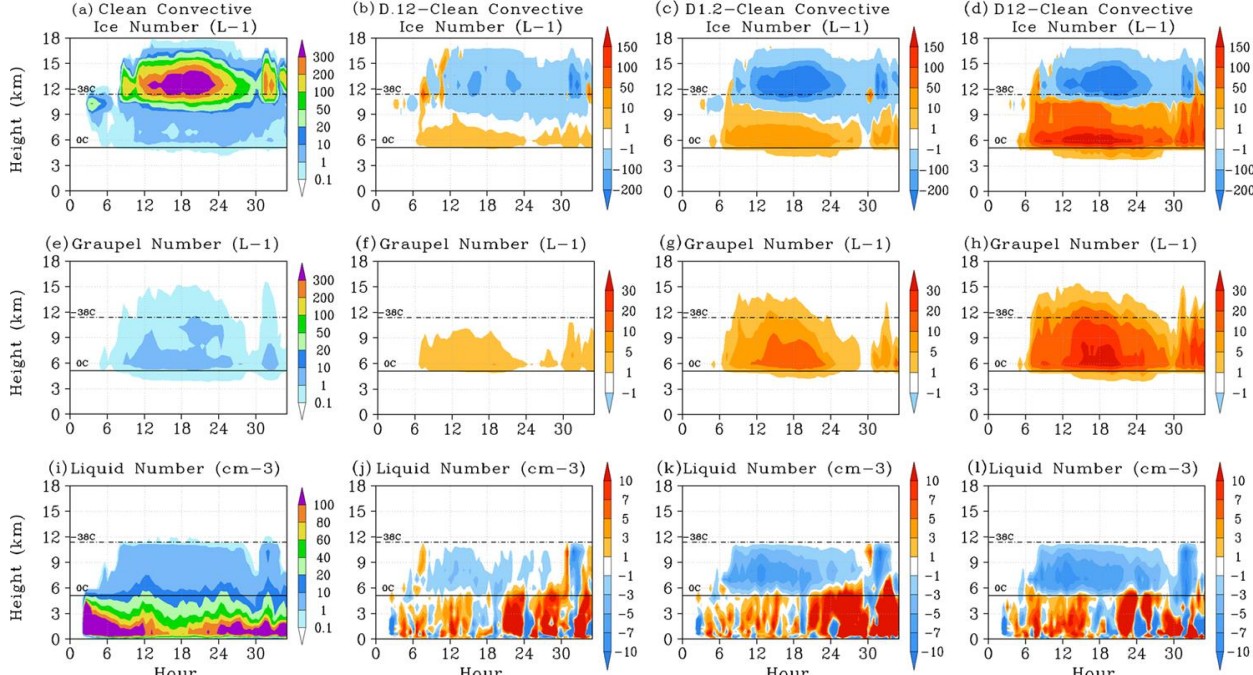

**Figure 8:** Convective cloud averaged profiles (Height vs time) and dust case minus Clean case difference plots of ice number concentration (top row), graupel number concentration (middle row), and liquid number concentration (Bottom row). Columns: Clean (a,e,i), D.12 - Clean (b,f,j), D1.2 - Clean (c,g,k), and D12 - Clean (d,h,l) cases.





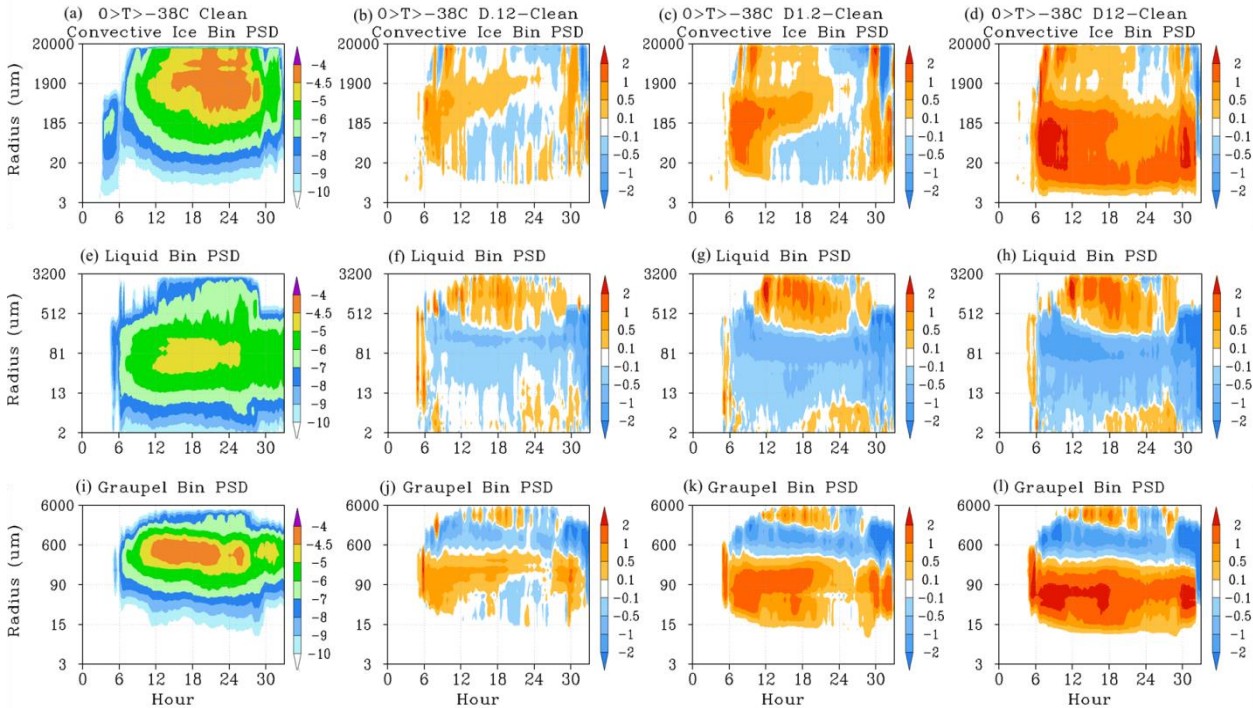

**Figure 9:** Time series and dust case minus Clean case difference plots of: ice/snow bin particle size distribution (PSD; top row), liquid bin PSD (middle row), and graupel bin PSD (Bottom row); averaged over the convective regime in the temperature range of -5°C to -38°C. Contours represent $\log_{10}$ values of bin population. Columns: Clean (a,e,i), D.12 - Clean (b,f,j), D1.2 - Clean (c,g,k), and D12 - Clean (d,h,l) cases.



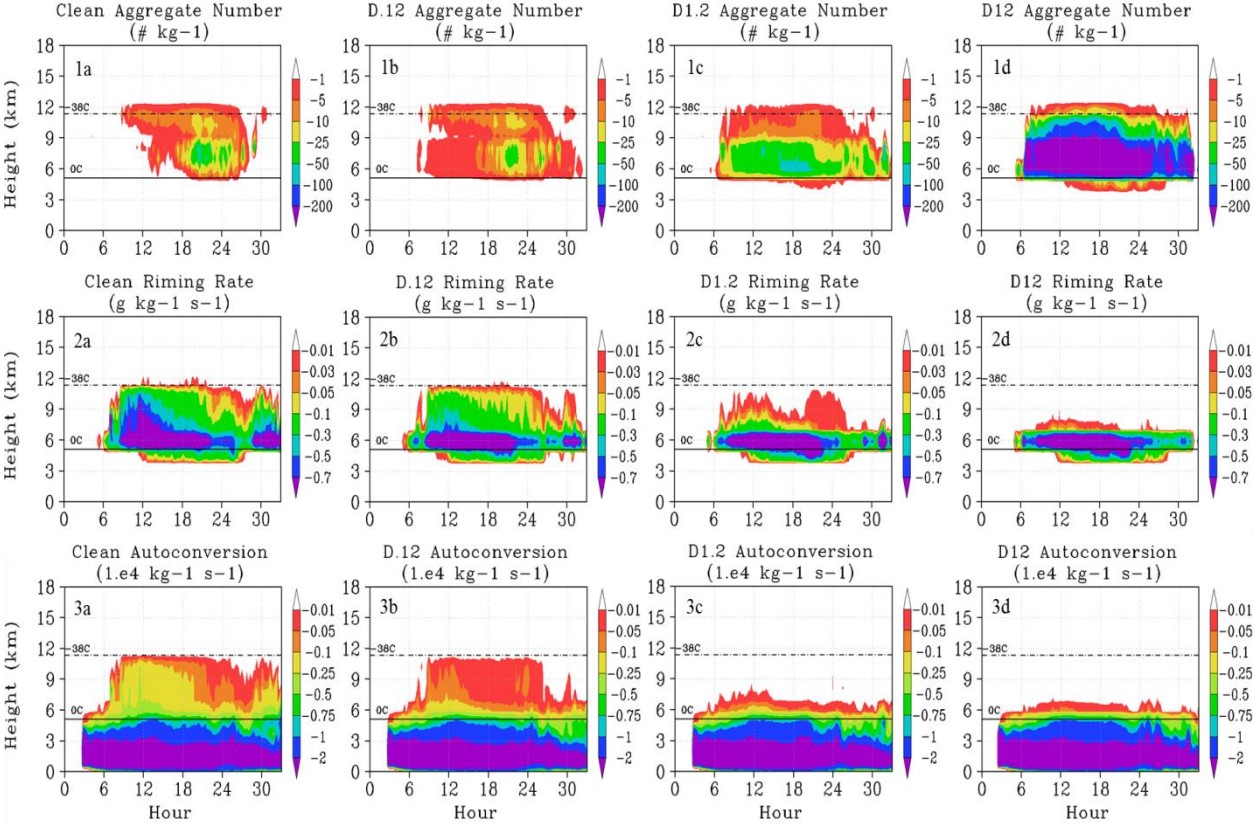

**Figure 10:** Time series of convective averaged aggregate number (row 1), riming rate (row 2), and drop autoconversion (collision-coalescence) number. Columns: Clean, D.12, D1.2, and D12 cases, respectively.




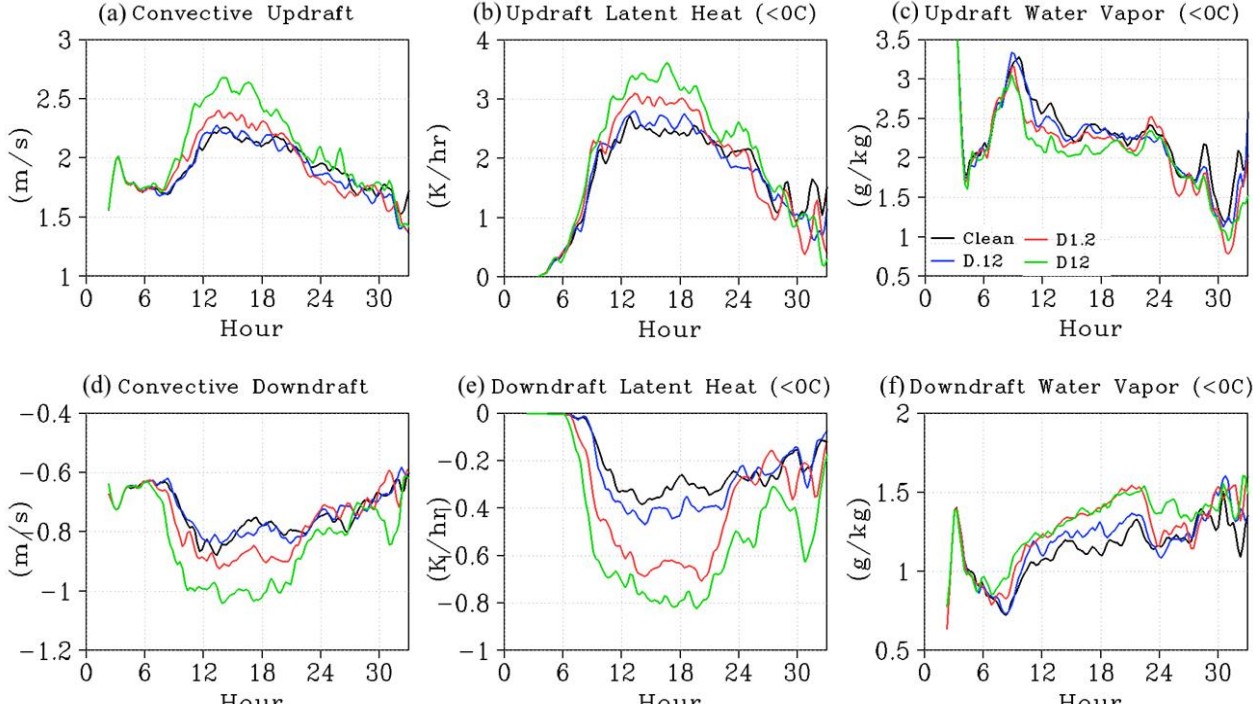

**Figure 11:** Top row: time series of average convective updraft intensity. Time series of average latent heat (K/hr) within convective updrafts (<0°C). Time series of average water vapour content (g/kg) within convective updrafts (<0°C). Bottom row: as top row, averaged over convective downdrafts. Colours represent Clean (black), D.12 (blue), D1.2 (red), and D12 (green) cases, respectively.


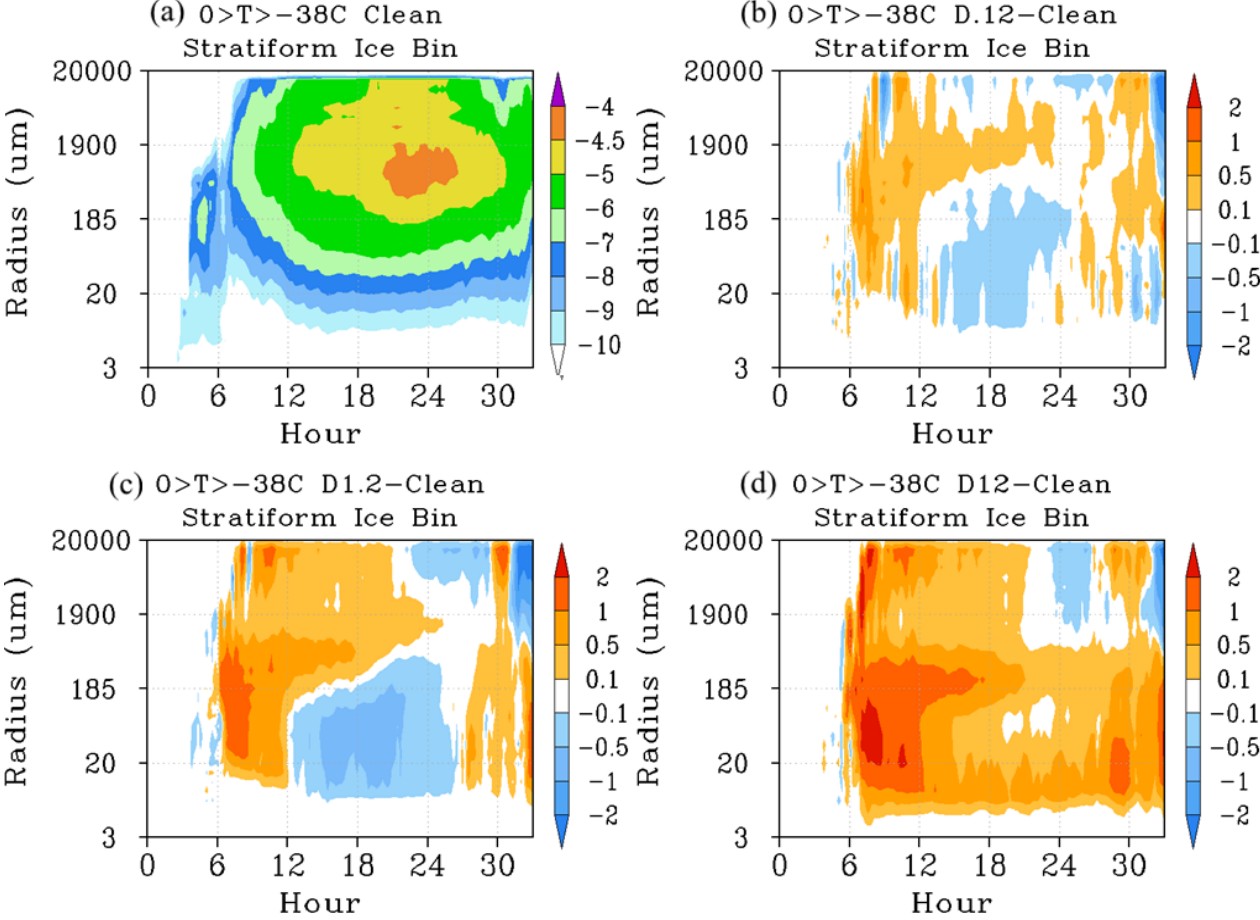

**Figure 12:** Time series and dust case minus Clean case difference plots of: ice/snow bin particle size distribution (PSD); averaged over the stratiform regime in the temperature range of -5°C to -38°C. Contours represent $\log_{10}$ values of bin population.



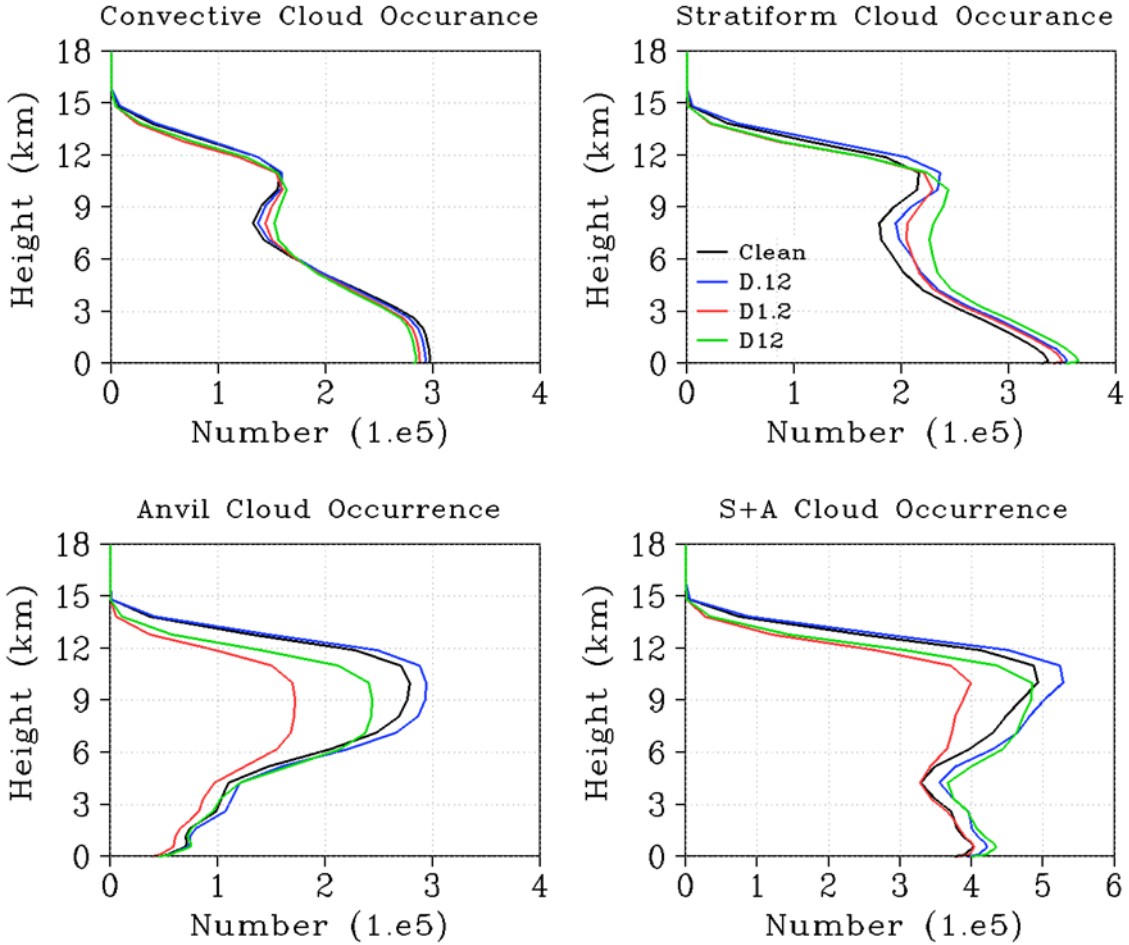

**Figure 13:** Occurrence frequency of cloudy data points over total simulation time for convective (a), stratiform (b), anvil (non-precipitating, c) and stratiform plus anvil (S+A, d) clouds. Colours represent Clean (black), D.12 (blue), D1.2 (red), and D12 (green) cases, respectively.



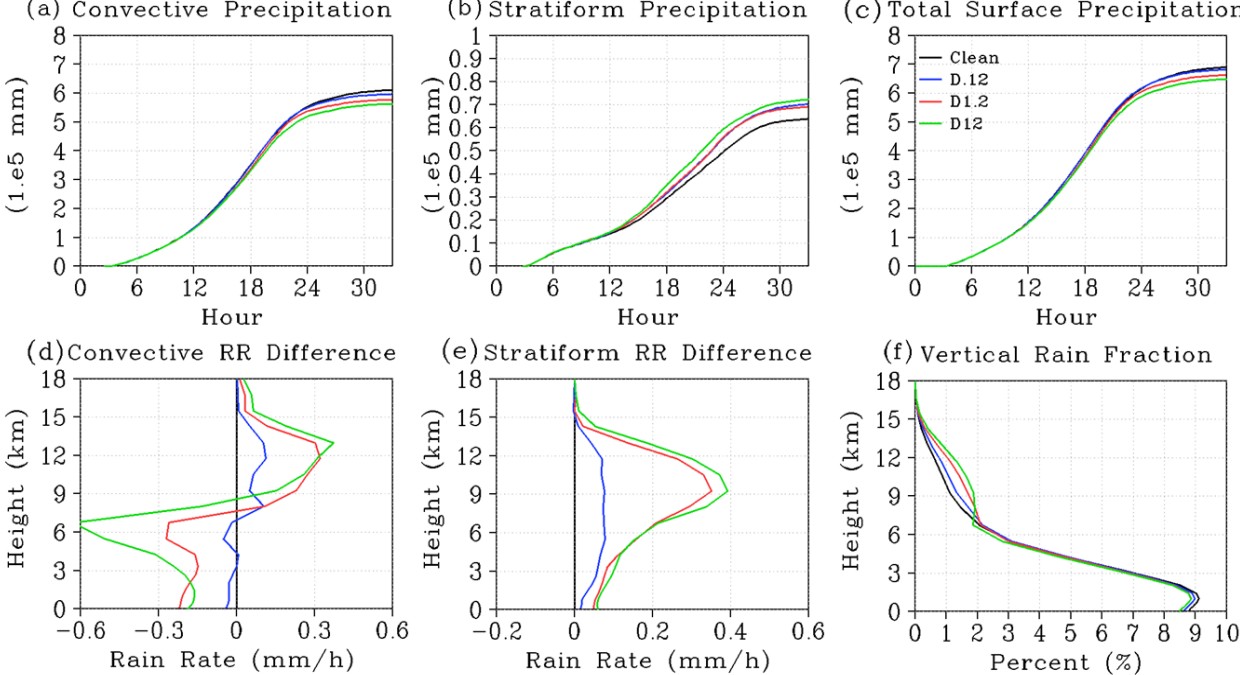

**Figure 14:** (a) and (b) Time series of accumulated surface precipitation for convective and stratiform data respectively. (c) Total accumulated surface precipitation for the clean and dust cases. (d) and (e) dust case minus Clean case time averaged vertical rain rates for convective and stratiform precipitation, respectively. (f) Fraction of total precipitation formed at each vertical level. Colours represent: Clean (black), D.12 (blue), D1.2 (red), D12 (green) cases, respectively.