# Peer review of "Investigating the Impacts of Saharan Dust on Tropical Deep Convection Using Spectral Bin Microphysics"

_Atmospheric Chemistry and Physics, 2017_

## Referee Comment (RC1) · Anonymous Referee #1 · 9 Sep 2017

Review of: Investigating the Impacts of Saharan Dust on Tropical Deep Convection Using Spectral Bin Microphysics. This is an interesting paper which examine the role of dust particles on deep convective clouds by acting as IN. To do so, the authors used the WRF model in a "real case" configuration coupled with bin microphysics scheme. The topic is of great importance and the tools used here are appropriate for making progress in understanding of it. However, I do have questions and suggestions for the authors: General comments I fill that more details are need in order for the reader to be able to fully evaluate the methodology:

Do you consider removal of IN by precipitation? Dry deposition? Do you consider

regeneration upon evaporation? Is the domain mean IN concentration constant with time? In addition, I guess that you don't divide the INs to bins, so how do you consider their sizes? What is their fall velocity?

The same goes for the CCN concentration, do you consider wet deposition? Regeneration upon evaporation? During 33 h of simulation with strong rain rates I guess the CCN concentration could change dramatically.

Regarding the model resolution, many previous studies have shown sensitivity to the resolution in respect to cloud resolving simulations. For example, Lebo and Morrison (2015) showed that only in ≥250m resolution the deep convective clouds' cores are well resolved. I suspect that 3km resolution for the inner domain is too coarse. Do you really need the outer domain to cover such a large area? I suggest to compromise on the total area covered by the simulation and go to higher resolution. Maintaining the same number of grid point but for smaller area won't increase the computation cost.

In many parts of the paper you repeat yourself (I listed a few examples blow but there are many more). Moreover, I think that with a bit of editing work this paper can become shorter and clearer, maintain its main results.

Specific comments

P2 L17 and 30: you say that dust particles are effective CCNs but later you describe totally different effect of dust on deep convective cloud than reported previously. How can it be integrated?

P3 L15-19: This sentence is unclear to me. What is the contradiction here? Since the 50's people attributed aerosol effect to changes in droplet size distribution.

P6 L7: How do you determine the rate of IN resupply?

P8 L1: Isn't it also proportional to the IN size?

P8 L23: How accurate those assumptions?

P13 L6: Here you declare stronger outflows from the core to the anvil, but in the abstract you wrote: "fewer particles form within and/or are transported into the anvil regime." Isn't it contradiction?

P13 L12: Suggest replacing "dynamical intensity" by "meteorological/environmental conditions" since dynamics and microphysics could be coupled.

P13 L14-16: It is an exact repetition. Pleas delete it.

P13 L29-31: Do you refer to the dusty case here? If yes, it will be nicer to spell this out.

P14 L12: I suggest referring here to two resent papers discussing the effect of aerosol on the vertical transport of hydrometers: How do changes in warm-phase microphysics affect deep convective clouds? Chen et al., 2017, ACP; Aerosol effect on the mobility of cloud droplets‏. Koren et al., 2015, ERL.

End of P14 and beginning of P15: It is repetition from above.

P16 L3: Suggest mentioning hear that the aerosol effect on clouds also depends on the environmental conditions and on the range of aerosol concentration examined.

P17 L29: It is repetition from above.

P18 L15: How do you define rain drops in the bin scheme?

P19 L14-17: Repetition from above.

Figure 11 c and f: Why not presenting super saturation (or saturation deficit) instead of water vapor concentrations? Since the temperature varies as well (different latent heat fluxes) the water vapor does not provide all necessary information.

P20 L6 and L10: Here some measure of the variance is needed to evaluate if order of 100m change in mean cloud top is significant.

P20 L21: Isn't it also because of the reduction in the available liquid water for freezing?

P21 L15, and L16-17: Repetition.

P22 L17: Is ∼1% change in precipitation significant compare to the simulations noise? In some cases different realizations of the same conditions could have higher different than that.

P22 L19: How do you determine that it is "significantly"?

P23 L4: "In the first of a two part study". This is the first time you mention another part of this study. What is the second part about?

P23 L18-19: The reasoning here is not clear to me. Is the stronger evaporation of the smaller hydrometers results in cooling and stronger downdrafts or vice versa (stronger downdrafts drive stronger evaporation)?

Technical comments

P8 L6: correct (m-3)

I suggest to have continues counting to the equations to avoid confusions.

P10 L20, L27, L29 and many other places: change "number" to "number concentration".

P12 L34: What does "1.e2 um" stand for?

P13 L28: Please correct: "th is".

I suggest organizing the figures in the order they are mentioned in the paper.

P19 L16: since only one case study is simulated here change "case studies" to "case study".

---

## Referee Comment (RC2) · Anonymous Referee #2 · 11 Oct 2017

**General comments**

Gibbons et al simulate an observed case of tropical deep convection under the influence of a dry and dusty Saharan Air Layer with spectral bin microphysics. The detailed cloud microphysical information is used to emulate the radar reflectivity from the model output. When correcting for a wet bias in the driving reanalysis data, the simulated radar reflectivites are found to be in accordance with the observations. With this observationally validated modeling framework, the authors investigate the effect of IN perturbations on the distribution of total cloud water into different hydrometeor classes and the corresponding changes in latent heat release. The microphysical re-

distributions are discussed in terms of the changes in cloud top height, precipitation and radar reflectivity that they correspond to. The paper constitutes a solid case study contribution but its presentation needs to be improved.

**Specific comments**

- The current manuscript seems to be a somewhat hasty merge of what was originally intended to be two companion papers. This is especially evident in the order of the figures and their referencing (for example, the text jumps from fig2 to fig7), and in the conclusion, which still contains the phrase "in the first of a two part study". This makes the manuscript in its current form hard to read such that it should be revised, especially in terms of reducing redundancies.

- The microphysical redistributions of cloud water between different hydrometeor classes and precipitation formation pathways in response to IN perturbations described in the manuscript should be discussed in the context of the existing literature, including literature on IN effects on other cloud types like cirrus and orographic clouds.

- The authors mention microphysical as well as thermodynamic effects on cloud height. Since this is a fundamental question in understanding convective invigoration, it would be desirable to put more focus on discussing the competition of the two mechanisms.

**Technical corrections**

1. p4 l4f: Do van den Heever et al (2006), Ekman et al (2007) and Tao et al (2012) consider prognostic IN?

2. section2: Are simulations performed with saturation adjustment or interactive supersaturation?

3. eq1: The definition of $F_M$ is not completely clear and it might help to define it using an equation.

4. fig3,5: Difference plots instead of the direct plots would be helpful.

---

## Author Comment (AC1) · 7 Dec 2017

Please see supplementary PDF for response

Please also note the supplement to this comment:
https://www.atmos-chem-phys-discuss.net/acp-2017-616/acp-2017-616-AC1-supplement.pdf

---

## Author Comment (AC2) · 7 Dec 2017

**General comments**

Gibbons et al simulate an observed case of tropical deep convection under the influence of a dry and dusty Saharan Air Layer with spectral bin microphysics. The detailed cloud microphysical information is used to emulate the radar reflectivity from the model output. When correcting for a wet bias in the driving reanalysis data, the simulated radar reflectivites are found to be in accordance with the observations. With this observationally validated modeling framework, the authors investigate the effect of IN perturbations on the distribution of total cloud water into different hydrometeor classes and the corresponding changes in latent heat release. The microphysical redistributions are discussed in terms of the changes in cloud top height, precipitation and radar reflectivity that they correspond to. The paper constitutes a solid case study contribution but its presentation needs to be improved.

**We thank the referee for their comments and suggestions to improve our manuscript. Author responses and changes are provided in bold text below.**

**Specific comments**

• The current manuscript seems to be a somewhat hasty merge of what was originally intended to be two companion papers. This is especially evident in the order of the figures and their referencing (for example, the text jumps from fig2 to fig7), and in the conclusion, which still contains the phrase "in the first of a two part study". This makes the manuscript in its current form hard to read such that it should be revised, especially in terms of reducing redundancies.

**We thank the referee for these suggestions. We have removed out of order figure references and modified any associated text in the corresponding analysis to improve the flow of the manuscript.**

**We have removed any mention of a second part and have removed redundant text wherever appropriate.**

• The microphysical redistributions of cloud water between different hydrometeor classes and precipitation formation pathways in response to IN perturbations described in the manuscript should be discussed in the context of the existing literature, including literature on IN effects on other cloud types like cirrus and orographic clouds.

**Aerosol indirect effects such as those resulting from IN perturbations have been more extensively studied in shallow clouds such as cirrus and orographic clouds due to their relatively simple mechanics compared to deep convective clouds. As such clouds do not feature the complex interactions between ice, liquid and mixed phase processes that occur in Deep convective clouds we focused our discussion on the observed MCS system. However, the effects of IN on shallow clouds has be recently reviewed by Fan et al. 2016.**

**We have added the following sentence after P2,L12: "AIE on shallow clouds has been extensively studied in the past as noted in the review by Fan et al. (2016), but additional research on AIE within deep clouds is still needed. The greater area coverage and lifetime persistence of the anvil cloud compared to the convective core makes the anvil cloud, and any changes resulting from AIE, important to global energy balance and radiative transfer. This makes the study of deep convective clouds important for current and future climate research (Solomon et al., 2007; Rosenfeld et al., 2013)."**

• The authors mention microphysical as well as thermodynamic effects on cloud
height. Since this is a fundamental question in understanding convective invigoration,
it would be desirable to put more focus on discussing the competition of
the two mechanisms.

**We agree that competition between the thermodynamical and microphysical effects are fundamental in understanding AIE. Our current study is intended to focus on numerically simulating the large scale effects of Dust on the MCS as described in Min et al 2009 and the associated studies. We have additional studies planned for the near future that will focus on the feedbacks between IN microphysical effects and latent heat processes in the DCC. This will allow for the more local interaction between thermodynamical and microphysical effects to be explored in greater detail.**

**We have added the following after P24,L26 to note this future work: "Additional in-depth study of the interactions between dust related microphysical effects and changes to latent heat processes has been planned for the near-future to more fully address the interconnected nature of thermodynamical and microphysical effects occurring within DCC."**

*Technical corrections*

1. p4 l4f: Do van den Heever et al (2006), Ekman et al (2007) and Tao et al (2012)
consider prognostic IN?

**van den Heever et al (2006) and Ekman et al (2007) do include prognostic calculation of IN concentration but do not connect IN to the extensive range of heterogeneous ice formation mechanisms that we have available in this study.**

2. section2: Are simulations performed with saturation adjustment or interactive supersaturation?

**The SBM calculates supersaturation interactively. Extensive details on the mechanics of the SBM were covered in Khain et al. (2004) and Fan et al. (2012a) and were not repeated in order to limit the length of our manuscript.**

3. eq1: The definition of $F_M$ is not completely clear and it might help to define it
using an equation.

**To clarify the definition, we have added the sentence following P7,L16: "For other values of ice supersaturation, $F_M$ is equal to $N_{id}$ divided by $N_{id}(S_i=40\%)$."**

4. fig3,5: Difference plots instead of the direct plots would be helpful.

**We have substituted difference plots as suggested and updated the relevant figure captions. Updated figures follow:**

[revised manuscript text omitted]

---

## Author Comment (AC4) · 21 Jan 2018

Please see supplementary file for Author's response.

Please also note the supplement to this comment:
https://www.atmos-chem-phys-discuss.net/acp-2017-616/acp-2017-616-AC4-supplement.pdf

---

## Referee Report (RR1)

**Review of: "Investigating the Impacts of Saharan Dust on Tropical Deep Convection Using Spectral Bin Microphysics" by Matthew Gibbons, Qilong Min and Jiwen Fan**

The revised manuscript is organized and written in a better and clearer way. In addition, the necessary details about the methodology were added. I have additional comments and suggestions for the authors.

**Specific comments**

I suggest to shorten the abstract and to focus only on the main conclusions here.

P3 L15: the increased evaporation of the smaller cloud droplet at the cloud periphery under polluted conditions can also increase the mixing between the cloud and the environment.

P14 L5: is it the same magnitude of forcing you get in the model results?

**Technical comments**

P2L12 (and other places): I suggest to be consistent and use "DCC" for "deep convective clouds" in all places.

P6 L19: at this point you still didn't mention the use of a nested grid with 4 domains so the reader don't know what does "the entire 4th domain" is referring to.

P10 L30: what is the scale of the exponential decrease in aerosol concentration with height?

P11 L2: you are talking here about wind shear but present values of wind speed.

P12 L28: "DCC cloud", you can delete the second "cloud".

P14 L23: correct: "From model hours 20 onwards, The…"

P19 L12 and P19 L25: change number to concentration − aggregation is sensitive to concentration and not to absolute number of partials.

P23 L14: suggest to add "(SBM)" after "spectral-bin microphysical"

I noted that you are referring to Fig. 6 only at the last paragraph of the manuscript. Isn't it appropriate to refer to it at the end of P15 or begging of P16?

Figures: correct cm-3 to $cm^{-3}$, L-1 to $L^{-1}$ and so on.

---

## Author Response (AR2)

We thank all reviewers for their time and critical review of the manuscript which we feel provided an important perspective on the material presented. As such, the manuscript is much more focused and streamlined. Our point-to-point responses to the reviewer are given below. For clarity, all responses are provided in blue

**Response to report #1**

I suggest to shorten the abstract and to focus only on the main conclusions here. The abstract is modified as suggested

P3 L15: the increased evaporation of the smaller cloud droplet at the cloud periphery under polluted conditions can also increase the mixing between the cloud and the environment. "Increased evaporation of smaller drops can result in stronger cold pool formation and enhanced secondary convection" is changed as suggested to "Increased evaporation of smaller drops can result in stronger cold pool formation and enhanced secondary convection, and such increased evaporation at the cloud periphery under polluted conditions can also increase the mixing between the cloud and the environment"

P14 L5: is it the same magnitude of forcing you get in the model results? The maximum difference in the model run is about  $40W/m^2$

P2L12 (and other places): I suggest to be consistent and use "DCC" for "deep convective clouds" in all places.

All deep convective clouds in the content has been changed into DCC except for when it first appeared.

P6 L19: at this point you still didn't mention the use of a nested grid with 4 domains so the reader don't know what does "the entire 4th domain" is referring to. The "4th domain" has been changed into a "certain domain" to avoid this problem.

P10 L30: what is the scale of the exponential decrease in aerosol concentration with height? The CCN concentration is calculated as followed: When height  $\leq 2km$ , CCN concentration = a certain number (300/cm3 in this case) When height >2km, CCN concentration =  $300*exp(-\frac{height-2km}{2km})$  (/cm3)

P11 L2: you are talking here about wind shear but present values of wind speed. (P11L20) "Wind shear" is corrected to "wind speed"

P12 L28: "DCC cloud", you can delete the second "cloud". Second "cloud" deleted

P14 L23: correct: "From model hours 20 onwards, The..." Corrected

P19 L12 and P19 L25: change number to concentration – aggregation is sensitive to

concentration and not to absolute number of partials. Number at those two places have been changed into concentration as suggested.

P23 L14: suggest to add "(SBM)" after "spectral-bin microphysical" SBM has been added as suggested.

I noted that you are referring to Fig. 6 only at the last paragraph of the manuscript. Isn't it appropriate to refer to it at the end of P15 or begging of P16? Yes. The description is for figure 6 but it wrongly quoted figure 5 at the beginning of p16. (Now figure 14 after changing the order of Results)

Figures: correct cm-3 to cm-3, L-1 to L-1 and so on. All corrections have been made in updated plots, as suggested.

**Response to report #2**

- The authors explain that (e.g. p23, l27/28) "Latent heat release in the heterogeneous nucleation regime is increased in the dust cases due to [..] smaller, more numerous particles". I do not think that this really is an effect of particle size but rather a consequence of an increased glaciation of the cloud. I mean to say that the total amount of ice, independent of whether it is distributed to many small or a few larger particles, controls vapor deposition. Yes, the water mass that go through the phase change is what decides the amount of latent heat release, "Latent heat release in the heterogeneous nucleation regime is increased in the dust cases due to diffusional growth and liquid-to-ice phase changes during riming of the smaller, more numerous particles." was changed into "In the dust cases, stronger diffusional growth and liquid-to-ice phase of latent heat release in the heterogeneous nucleation numerous particles." to better describe the process.

- I am struggling to reconcile Figs. 10 and 14 (d) with the authors conclusion on "less efficient graupel formation reducing convective rain rates" (e.g. p1, l24): Fig 14 (d) show that the reduction is dominated by a reduction around 6km height. In Fig. 10, riming seems to be about the same, while autoconversion is decreased and aggregation increased in this region - so for me it seems that a reduction in autoconversion is the cause of decreased precipitation. Could you clarify this, maybe with the help of difference plots in Fig. 10? Similarly, do you provide an explanation for your observation that "precipitation formation is shifted to colder temperatures" (e.g. p24, l2)? Comparing Fig. 14 (d) and Fig. 10 this seems to correspond to a shift from autoconversion to aggregation.

As you pointed out, when extra IN is added to the system, the heterogeneous nucleation is enhanced, the system has less water to form liquid droplets and autoconversion will indeed decrease. However, from figure 10 (now figure 8) we can also see that the aggregation (ice collection) is becoming stronger as we increase the IN number. The reduction at ~6km is duo to both effects. The shift from autoconversion to aggregation results in the shift of precipitation formation to colder temperatures.

- I suggest to move section 4.2 "Radar Reflectivity" to the end of section 4. This would allow

to discuss its findings using results from the foregone analysis. Also, it would reflect the structure of abstract and conclusion. The order is adjusted as suggested

- The authors find that "reducing dust layer moisture content by 5% " (e.g. p25, 11) inverts the convective radar reflectivity difference pattern. What does this sensitivity to layer moisture mean for the robustness of the results?

Measurements from AIRS/AMSU/HSB indicate that the relative humidity in the dust layer is about 20% drier than the surrounding air. The sensitivity study with reduced moisture content is to mimic this observed conditions.

**Smaller/technical comments:**

- p1, 120: Do you mean: Before they are transported into the anvil? Yes.

- p6, 119: The "4th domain" is mentioned here without having been introduced. The "4th domain" has been changed into a "certain domain" to avoid this problem.

- p7, 18: The sentence does not make sense as is.

"Currently there is no deposition and condensational nucleation parameterization connecting with aerosol properties and developed based on deep convective clouds" is changed into "Currently there is no deposition and condensational nucleation parameterization that is developed to connect with aerosol properties for DCC."

- p13, 113: Where is the rain rate increased? Near the equator? Below the core it seems decreased?

The rain rate below 0  $^{\circ}$ C (at higher altitude than the 0  $^{\circ}$ C layer) has increased, as well as the reflectivity.

- p13, l21: Could you also comment on the strong increase in snow radius below the freezing level?

Similar to graupel, the snow radius is increased due to immersion freezing of large rain drops

- p17, 11: The black lines in Fig. 7 (a,b) do not look like more than 90% of ice formation occur above the -38 degree line. Could you clarify this?

Because that ice can be transported by air flow, ice found above -38  $^{\circ}$ C layer is not necessarily produced there. Percentage of homogeneously formed ice at above -38  $^{\circ}$ C layer is reducing with increasing IN number. The sentence is indeed confusing and is deleted.

- p 21, 116-19: I find this very hard to see - could you show/discuss this more clearly? The convective PSD is evolving similarly to the stratiform PSD. From hour 6 when the clouds begin to develop to hour 18 when the clouds reach their mature stage, we can see that compared with the clean case, there is a tendency in the D.12 and D1.2 cases that from hour 12, the number of small ice particles is keep decreasing while the number of large particles is increasing, and the radius threshold between large and small is increasing. That is, the number of large particles is increasing and the average size of large particles is becoming larger. However, after hour 24, we can see from figure 7a and figure 9a that the clouds themselves are becoming weaker, number of particles of all size are decreasing. The large particles fall out first, therefore the reversed pattern after hour 24.

- p24, l2 and l 17/18: These sentences repeat each other. L2 was deleted.

- Fig 2: Why don't you show difference plots here? In figure 2, the different effect of extra IN on the pattern and intensity are quite substantial especially above 6km. The difference is clear to see in those plots. Using difference plots may not make them clearer.

- Fig 10: Why don't you use difference plots here? Can aggregate numbers be converted to rates so that there magnitudes can be compared to riming and autoconversion? Autoconversion rates in the warm region exceed the color scale for all dust levels so that they cannot be compared. Could you add a level to the color scale or use difference plots? Similar to the previous reply, using difference plots may not make points clearer. We tried to add a level to the color scale, but it did not show significant difference.